# Broaden your SCOPE! Efficient Multi-turn Conversation Planning for LLMs using Semantic Space

**Zhiliang Chen**[1,2,*]**, Xinyuan Niu**[1,3,*]**, Chuan-Sheng Foo**[2,3] **& Bryan Kian Hsiang Low**[1]

[1]Department of Computer Science, National University of Singapore
[2]Institute for Infocomm Research (I2R), A*STAR, Singapore
[3]Centre for Frontier AI Research (CFAR), A*STAR, Singapore
`{chenzhiliang, xinyuan}@u.nus.edu`
`foo_chuan_sheng@i2r.a-star.edu.sg`
`lowkh@comp.nus.edu.sg`

## Abstract

*Large language models* (LLMs) are used in chatbots or AI assistants to hold conversations with a human user. In such applications, the quality (e.g., user engagement, safety) of a conversation is important and can only be exactly known at the end of the conversation. To maximize its expected quality, conversation planning reasons about the stochastic transitions within a conversation to select the optimal LLM response at each turn. Existing simulation-based conversation planning algorithms typically select the optimal response by simulating future conversations with a large number of LLM queries at every turn. However, this process is extremely time-consuming and hence impractical for real-time conversations. This paper presents a novel approach called **S**emantic space **CO**nversation **P**lanning with improved **E**fficiency (SCOPE) that exploits the dense semantic representation of conversations to perform conversation planning efficiently. In particular, SCOPE models the stochastic transitions in conversation semantics and their associated rewards to plan entirely within the semantic space. By doing so, SCOPE selects the optimal LLM response at every conversation turn without needing additional LLM queries for simulation. As a result, SCOPE can perform conversation planning *70 times faster* than conventional simulation-based planning algorithms when applied to a wide variety of conversation starters and two reward functions seen in the real world, yet achieving a higher reward within a practical planning budget. Our code can be found at: `https://github.com/chenzhiliang94/convo-plan-SCOPE`.

## 1 Introduction

The rise of *large language models* (LLMs) has introduced numerous conversational tools in the market, such as Replika (`https://replika.com`) and chatbots (Dam et al., 2024). As these conversational tools become widespread, there are significant commercial and regulatory interests in ensuring that they produce high quality conversations with the human users. For example, a chatbot is commercially motivated to select responses at every conversation turn to produce a longer and more engaging conversation with a human user. In this case, the quality of a conversation can only be exactly known at the end of the conversation by accumulating the rewards (according to some metric such as conversation engagement, safety) over multiple turns. To produce an *optimal conversation* where the cumulative reward across multiple turns is maximized, we need to perform *conversation planning* that reasons about the stochastic transitions in a conversation to strategically select an optimal LLM response at each conversation turn.

Several recent works are *myopic* in conversational planning because they merely select the LLM response that seems immediately "good" at each conversation turn. For example, one might be tempted to use a predefined reward function to select the LLM response with the highest immediate reward

---

*Equal contribution.

(Kumar et al., 2024; Yang et al., 2023) at each turn. However, myopic approaches do not consider how the selected LLM response influences the conversation (and rewards) later on. This implies a response that seems good at first might not lead to better conversation rewards. In Appendix A.1, we give some illustrative examples where LLM responses with similar instantaneous rewards eventually lead to conversations of differing quality in metrics such as conversation harmfulness. Hence, without explicitly reasoning about how a selected LLM response influences the conversation later, myopic approaches are suboptimal in maximizing the quality of the overall conversation.

Alternatively, an LLM can be fine-tuned (Ouyang et al., 2022; Rafailov et al., 2023) to produce a response that implicitly maximizes a cumulative reward metric across multiple conversation turns. While non-myopic if done correctly, these approaches are resource-intensive (Zhang et al., 2024), requiring manual preference labels and repeated model fine-tuning. Therefore, fine-tuning approaches are impractical for an LLM owner deploying LLMs for different use cases that require different reward metrics. On the other hand, our paper focuses on the *training-free* setting (no fine-tuning of LLMs) and explores inference time strategies (Lin et al., 2024; Zhou et al., 2024b; Wu et al., 2024; Xu et al., 2024) to select LLM responses in a conversation.

More recently, some works have also explored using LLMs and simulation-based search algorithms (Yu et al., 2023; Koh et al., 2024; Hao et al., 2023; Kim et al., 2024) to plan in LLM-related problems such as goal-oriented dialogues (Yu et al., 2023), language games (Jang et al., 2021) and more. The popularity of these algorithms is driven mostly by the ubiquitous effectiveness of LLMs, using LLMs to simulate conversation users (Liu et al., 2023) to find out which responses lead to higher cumulative rewards in the simulated conversation. However, these simulation-based algorithms suffer from high planning costs (Koh et al., 2024; Zhao et al., 2023), requiring *hundreds or even close to thousands of seconds* to perform sufficient look-ahead simulation during runtime (details covered in Section 3.2). Such large planning budget is a luxury unavailable in real-time conversations, where a human user expects the LLM to respond almost immediately. Thus, given practical planning budget, these simulation-based search algorithms do not achieve sufficient simulation scope to select optimal LLM responses. Furthermore, these LLM queries could also incur excessive monetary costs if they use API calls from external LLM providers. Therefore, conventional simulation-based algorithms are impractical in our problem setting due to the large planning costs involved. In our experiments (Section 6.3), we show that such algorithms indeed perform poorly in conversation planning under practical planning budget.

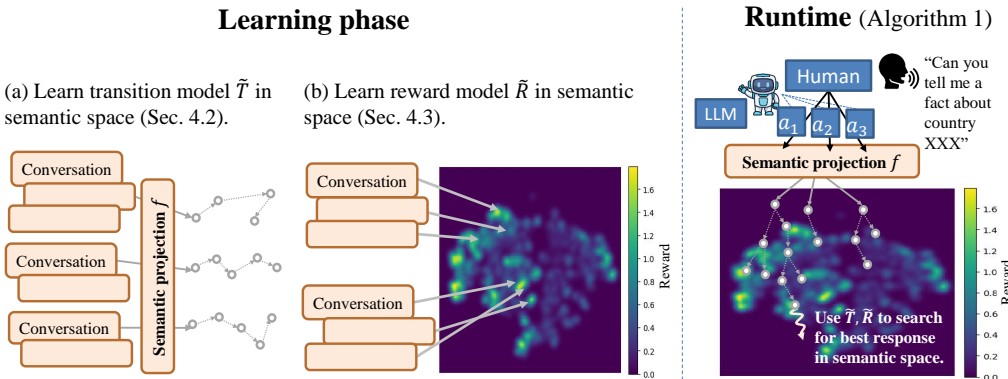

Figure 1: Overview of our approach, SCOPE, for conversation planning. During learning phase (left), we exploit dense semantic representation of conversations to learn a lightweight transition and reward model in semantic space (Notations and details covered in Sections 4.2 and 4.3). During runtime (right), SCOPE uses these learnt models to perform MCTS at a broader scope in semantic space without needing additional LLM queries for simulation (Algorithm 1), allowing us to select an optimal LLM response that maximizes the cumulative reward in a conversation.

This paper presents a novel approach called **S**emantic Space **CO**nversation **P**lanning with improved **E**fficiency (SCOPE) which exploits the dense semantic representation of conversations to perform conversation planning in a *semantic space* (Figure 1). Such representations have been shown to capture the semantics of natural language conversations effectively (Devlin et al., 2019; Chen et al., 2020). During the learning phase, SCOPE learns the stochastic transitions in conversation and their

associated rewards using a lightweight transition and reward model in semantic space. During runtime, SCOPE uses the learnt models to perform Monte Carlo tree search (MCTS) and simulation in semantic space, selecting an LLM response that leads to high quality conversations without using costly LLM queries. This results in a non-myopic approach that is effective even under realistic planning budgets. As such, our work presents a paradigm shift, similar to that found in Kambhampati et al. (2024): instead of relying on an LLM fully for simulation in conversation planning, our approach only uses an LLM to propose candidate responses, before using SCOPE to plan in semantic space without needing costly LLM queries. Concretely, our contributions are:

- We transform the multi-turn conversation planning problem, originally viewed as a *Markov decision process* (MDP), into its semantic form representation (Section 4.1). The semantic form preserves the optimal action (LLM response) selected at every conversation turn in a real-time conversation.
- We introduce a novel approach, SCOPE, that solves this transformed MDP by exploiting dense conversation semantic representations to learn a lightweight transition and reward model over conversation semantics. By modeling the stochastic transitions within a conversation and their associated rewards in semantic space, SCOPE performs MCTS entirely in semantic space without needing additional LLM queries to search for optimal LLM responses at every conversation turn. By doing so, SCOPE plans *70 times faster* than conventional simulation-based planning algorithms, allowing it to select optimal LLM responses under practical planning budget.
- We use SCOPE to select LLM responses in a wide range of real-world multi-turn conversations and show that learning the transition and reward models in semantic space allows SCOPE to achieve higher cumulative rewards than conventional non-myopic and myopic planning algorithms under practical planning budget and two realistic reward functions.

## 2 CONVERSATION PLANNING FOR LLMS

### 2.1 PROBLEM SETTING

Our work focuses on real-time, multi-turn conversations between the LLM and a human user. Briefly, a conversation consists of alternating responses given by each party at every conversation turn. Given a reward function which assigns a numerical reward to each conversation turn (dependent on the response given by the LLM and/or human user), *conversation planning* aims to select the optimal LLM response out of a candidate pool of responses at every conversation turn, maximizing expected cumulative reward over multiple

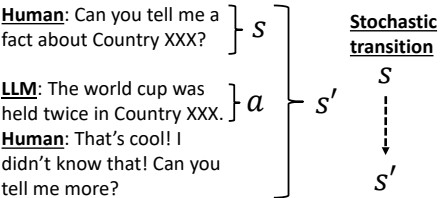

Figure 2: Example of a stochastic transition from $s$ to $s'$ after taking action $a$ in MDP for conversation planning.

turns. With an appropriately chosen reward function (some examples given in Section 6.1), the cumulative reward indicates the overall quality of the conversation (e.g., engagement, safety). Conversation planning is challenging because (a) we need to account for the stochastic response of the human user (which we cannot directly control) at every conversation turn and (b) an LLM response selected at the current turn influences later conversations, affecting future rewards obtained.

Inspired by similar challenges found in the broad class of *sequential decision making* problems (Bellman, 1957) and dialogue planning (Yu et al., 2023) containing stochastic environments and sequential action selection, we formulate conversation planning as a Markov decision process (MDP) (Howard, 2012) defined by $(S, A, T, R)$ shown in Figure 2, where

- $S$ is a set of states. Each state $s \in S$ is the conversation context comprising all responses given chronologically from prior conversation turns and ending in a human response.
- $A$ is a set of actions. Each action $a \in A$ is an LLM response shown to the human user at a conversation turn. Similar to prior works, a candidate set of LLM responses is generated at each turn in which $a$ is selected. After selecting an LLM response (i.e., taking action $a$), the human user responds, transitioning the conversation to a new state $s'$ (see Figure 2).
- $T : S \times A \times S \mapsto [0, 1]$ is a transition function specifying the probability of transitioning from a conversation state $s \in S$ to the next state $s' \in S$ after executing action $a$ (i.e., selecting a particular LLM response to show the human user). $T$ captures how a human user stochastically

responds at every conversation turn. In our paper, we are concerned about modeling a simulator to sample transitions from $T$ and do not necessarily need to calculate the transition probability.

- $R(s, a, s')$ is an instantaneous reward function associated with the state transition at every conversation turn. This reward can be dependent on both LLM and human response (i.e., dependent on $s$, $a$ and $s'$), and is chosen by the LLM owner beforehand.

Given this MDP, our goal is to, at each conversation state $s$, select an LLM response $a_s^*$ which maximizes the expected cumulative reward (w.r.t. the stochastic transition function) over a conversation horizon (e.g., 5 conversation turns):

$$a_s^* \triangleq \arg\max_{a \in A} \sum_{s' \in S} T(s, a, s')(R(s, a, s') + \gamma V(s')) \tag{1}$$

where $\gamma$ is the discount factor (Amit et al., 2020) and $V(s')$, the *value* of state $s'$, is recursively defined as

$$V(s') \triangleq \max_{a' \in A} \sum_{s'' \in S} T(s', a', s'')(R(s', a', s'') + \gamma V(s'')). \tag{2}$$

While one is tempted to solve our MDP using off-the-shelf approaches such as value iteration (Pashenkova et al., 1996) to learn an optimal response selection policy, these approaches are infeasible in our problem setting due to the tremendous number of possible conversations between a human user and the LLM (i.e., huge number of states and actions).

## 3 MONTE CARLO TREE SEARCH (MCTS) FOR CONVERSATION PLANNING

### 3.1 DESCRIPTION OF VANILLA MCTS

More recently, simulation-based planning algorithms like MCTS (Browne et al., 2012) have shown success in tackling MDPs surrounding language-based tasks (Yu et al., 2023; Koh et al., 2024) by planning around more promising actions and states. MCTS uses a simulator of the environment to explore future states and actions, using observed cumulative rewards from simulation to learn *state-action values* via a Q-function $Q(s, a)$ (Mnih et al., 2013; Watkins & Dayan, 1992). The state-action value approximates the expected cumulative reward that can be achieved by action $a$ at state $s$.

Our approach, SCOPE, uses MCTS as its backbone. Here, we provide a brief overview of MCTS (details can be found in Appendix A.2). In MCTS, we assume access to a budget of $K$ iterations. At iteration $k$ during the selection phase, we traverse a search tree from the starting state according to the Upper-confidence Tree (UCT) (Galván & Simpson, 2021) tree-search policy: $\pi(s) \triangleq \arg\max_{a \in A} \left( Q_k(s, a) + \lambda \sqrt{(\log N_k(s))/N_k(s, a)} \right)$ where $Q_k(s, a)$ is the state-action value estimate of selecting action $a$ at state $s$ during iteration $k$, $N_k(s)$ is the number of times $s$ is visited, $N_k(s, a)$ is the number of times action $a$ has been selected at $s$, and $\lambda$ is a constant which encourages exploration when selecting actions. With the ubiquitous success of LLMs, using an external LLM as a simulator of human user (Yu et al., 2023; Yao et al., 2024) has become a popular way to perform simulation rollouts in MCTS (although we argue in the next section that doing so is impractical). Finally, the state-action value is updated with observed cumulative reward $\hat{R}$ (with discount factor $\gamma$) at the end of iteration $k$ via $Q_k(s, a) = Q_{k-1}(s, a)(1 - 1/N_k(s, a)) + \hat{R}/N_k(s, a)$. From hereon, we refer to the procedure of MCTS with an external LLM simulator as **vanilla MCTS**.

### 3.2 BOTTLENECK FOR REAL-TIME CONVERSATION PLANNING

While MCTS has shown promising results in games such as Go (Silver et al., 2016) and Hearthstone (Świechowski et al., 2018), it is impractical for real-time conversation planning as it requires a large planning budget to simulate future conversations. In particular, performing multiple simulation rollouts at every conversation turn with an external LLM acting as a human user simulator requires a large number of LLM inferences (or API calls to an LLM service provider, such as ChatGPT (OpenAI, 2023)). Using an LLM to simulate a conversation with responses of reasonable length takes a few seconds. So, the key bottleneck is that we cannot perform sufficient number of simulation rollouts (a key step in the MCTS algorithm) within a reasonable amount of time. As we can only simulate a very narrow scope of search space, we cannot accurately estimate state-action values to select the optimal LLM response at every turn. Interestingly, we note that many prior simulation-based planning approaches for LLMs (Koh et al., 2024; Jang et al., 2021; Zhou et al., 2024a) use

large amount of time to plan effectively during runtime. In fact, results from Yu et al. (2023) showed that *740 seconds* of search time (at *each* conversation turn) is needed for vanilla MCTS to achieve reasonable planning performance in a goal-oriented dialogue task. This is clearly unreasonable for any real-time conversation constrained by a small window of response time. Indeed, our experiments (Section 6.3) show that vanilla MCTS is ineffective for conversation planning under realistic planning budgets. To overcome this bottleneck, we introduce SCOPE in the next section, a novel approach which exploits conversation semantics to avoid costly simulations with LLM queries.

# 4 INTRODUCING SCOPE

The key intuition behind SCOPE is to perform MCTS without explicitly simulating future conversations with costly LLM queries. To do so, we leverage dense semantic representations (Devlin et al., 2019; Lee et al., 2024; Rui et al., 2024; Lau et al., 2024a) of natural language in a continuous space which we refer to as *semantic space*. Notably, we found that a conversation can be captured effectively by a series of stochastic transitions in the semantic space (we provide empirical evidence in Appendix A.8) across multiple turns. By predicting these transitions and learning the instantaneous rewards associated with each point in semantic space, we can perform MCTS with simulations in semantic space instead of using time-consuming LLM queries. This results in a non-myopic approach that selects the optimal LLM responses in real-time conversations. While prior state abstraction works (Hostetler et al., 2014) have tried to cluster MDP states based on feature similarities, they still rely on costly simulations in the original environment. On the contrary, our approach performs simulations entirely in the semantic space, which are much less time-consuming.

SCOPE consists of two distinct phases (Figure 1). During the learning phase, we use static conversation data to learn a lightweight transition and reward model over the semantic space. Both models are significantly faster to query as compared to LLMs, as only a single forward pass of the model is required to simulate each conversation transition. The transition model (Section 4.2) predicts how a conversation transitions stochastically from one point to another in the semantic space while the reward model (Section 4.3) predicts the reward associated with each point in semantic space. Therefore, every conversation can be represented as a path in this semantic space, and our goal is to search for the optimal next immediate action that leads to a path with the highest cumulative reward. We provide several examples of such paths in Figures 3a and 3c. During runtime, SCOPE uses a semantic embedding model to project the conversation starter and candidate LLM responses into the semantic space. Then, SCOPE uses the learnt models from the learning phase to broaden the scope of search in semantic space (Algorithm 1) without any need for additional LLM queries. Our experiments (Section 6.3) show that under different planning budgets, SCOPE consistently achieves higher cumulative rewards than other baselines.

## 4.1 SEMANTIC SPACE REPRESENTATION

We use a semantic embedding model $f : S \mapsto \mathbb{R}^n$ to map conversation states $s$ into its semantic representation $\tilde{s} \triangleq f(s)$. In practice, we can use of-the-shelf text embedding models as $f$. With this, we rewrite the original MDP (1) for conversation planning into

$$\arg\max_{\tilde{a}} \sum_{\tilde{s}'} \widetilde{T}\left(\tilde{s}, \tilde{a}, \tilde{s}'\right)\left(\widetilde{R}(\tilde{s}, \tilde{a}, \tilde{s}') + \gamma \widetilde{V}(\tilde{s}')\right) \tag{3}$$

where $\widetilde{T}$ is the transition model over semantic space, $\widetilde{R}$ is the instantaneous reward associated with each conversation turn in semantic space, $\widetilde{V}$ is defined recursively similar to Equation (2), and $\tilde{s}'$ is the resulting state in semantic space after a transition. In this new MDP, state $\tilde{s}$ represents a *point* in semantic space, $\tilde{a}$ represents an action (possible LLM response) in the semantic space. A conversation's semantic transition $\tilde{s} \rightarrow \tilde{s}'$ is represented as a *directional vector* along a path shown in Figure 1. We theoretically show in Appendix A.3 that under some assumptions of $f$, $\widetilde{R}$ and $\widetilde{T}$, new MDP Equation (3) yields the same solution as the original MDP Equation (1). By learning $\widetilde{T}$ and $\widetilde{R}$ using static conversation data (details covered next), SCOPE performs MCTS in the $\mathbb{R}^n$ semantic space to solve MDP Equation (3) without needing costly LLM queries. In our experiments, we show that even if assumptions on $f$, $\widetilde{R}$ and $\widetilde{T}$ do not hold true, SCOPE consistently outperforms other baselines by selecting LLM responses that yield higher cumulative rewards.

## 4.2 Learning transition model $\widetilde{T}$ in semantic space

Given a semantic embedding model $f : S \mapsto \mathbb{R}^n$, our first step is to learn a transition model $\widetilde{T}(\tilde{s}, \tilde{a}, \tilde{s}')$, allowing us to simulate transitions in semantic space (we do not necessarily need to derive the transition probability explicitly). To do so, we make use of a conversation dataset containing $N$ transitions: $\{(s_1, s_1'), (s_2, s_2'), \ldots, (s_N, s_N')\}$, where $s_i, s_i'$ represents an observed transition from one conversation state to another. For example, the sentence pair: (1) "**Human**: Hello how are you?" and (2) "**Human**: Hello how how are you? **LLM**: I'm good, thank you! What did you do this weekend? **Human**: I went to the cinema!" represents a transition at one conversation turn ("I'm good, thank you! What did you do this weekend?" is an action $a$ taken by the LLM). Next, we use a semantic embedding model $f$ to transform this dataset into $\{(\tilde{s}_1, \tilde{s}_1'), (\tilde{s}_2, \tilde{s}_2'), \ldots, (\tilde{s}_N, \tilde{s}_N')\}$. Ideally, a learnt $\widetilde{T}$ should allow us to sample semantic transitions similar to those found in the dataset.

At first glance, one might treat this as a multi-output regression problem in a supervised learning setting (given a labeled dataset of conversation transitions) and learn a deterministic neural network $F_T(\tilde{s}, \theta_T) : \mathbb{R}^n \mapsto \mathbb{R}^n$ with network weights $\theta_T$ according to the mean squared error loss:

$$\min_{\theta_T} N^{-1} \sum_{i=1}^N \left( F_T(\tilde{s}_i, \theta_T) - \tilde{s}_i' \right)^2 \tag{4}$$

that approximates transitions from $\widetilde{T}$. However, this neural network, being deterministic, cannot simulate the stochastic nature of transitions within conversations. Instead, we can use probabilistic models such as mixture density network (MDN) (Bishop, 1994) or deep ensembles (Lepikhin et al., 2021) to model $\widetilde{T}(\tilde{s}, \tilde{a}, \tilde{s}')$ based on the labeled dataset. These probabilistic models are more suitable largely due to their abilities to draw samples from transition distributions (more details in Appendix A.6). We provide empirical evidence that these probabilistic models can indeed model stochastic semantic transitions well in Appendix A.8. While it is not immediately clear which probabilistic model choice of $F_T(\tilde{s}_i, \theta_T)$ leads to better performance in SCOPE, we perform ablation studies in Section 6.4 to tease out the influence of different transition model choices on performance of SCOPE. Note that in our implementation, $\widetilde{T}$ consists of two separate models, one to predict $\tilde{s} \to \tilde{a}$ and another to predict $(\tilde{s}, \tilde{a}) \to \tilde{s}'$ in semantic space (details in Appendix A.6).

## 4.3 Learning Reward model $\widetilde{R}$ in semantic space

In the original problem setting, it is easy to derive the instantaneous reward $R(s, a, s')$ of Equation (1) at each conversation turn. For example, if we want to minimize the number of harmful words uttered over the entire conversation, we can simply count the number of harmful words appearing at each conversation turn and treat that as the instantaneous reward during simulation in MCTS. However, as SCOPE performs MCTS in the semantic space, we cannot directly derive the instantaneous reward during simulation because we do not know the reward as we transition from one point to another in semantic space. To resolve this, we use another neural network $F_R(\tilde{s}, \theta_R)$, parameterized by $\theta_R$, to estimate the reward associated with every point $\tilde{s}$ in the semantic space. Then, for any state $\tilde{s}'$ encountered after performing $\tilde{a}$ at $\tilde{s}$ in semantic space, the instantaneous reward in semantic space $\widetilde{R}(\tilde{s}, \tilde{a}, \tilde{s}')$ (Equation (3)) can be recovered via $\widetilde{R}(\tilde{s}, \tilde{a}, \tilde{s}') \approx F_R(\tilde{s}', \theta_R) - F_R(\tilde{s}, \theta_R)$ (this is an approximation if the neural network cannot learn the rewards perfectly). We provide rigorous explanation on the validity of this recovery process and details of the training process for the reward model $F_R(\tilde{s}, \theta_R)$ from data in a supervised setting in Appendix A.4. Note that in practice, we can avoid training a reward model and directly use off-the-shelf models by using the same model for reward and semantic embedding (details in Section 6.2 and Appendix A.6).

## 5 Using SCOPE during runtime

As we show in our experiments (Section 6.3), the learning phase only needs to be conducted once to learn $\widetilde{T}$ and $\widetilde{R}$ before deploying them for different real-time conversations. During runtime, SCOPE uses the same learnt transition and reward models $\widetilde{T}, \widetilde{R}$ to perform MCTS (similar to that in Section 3) in semantic space. Algorithm 1 demonstrates SCOPE at every conversation turn.

Given a conversation context $s_{\text{init}}$ ending with a prompt from a human user, the LLM, like prior works (Kambhampati et al., 2024; Yang et al., 2024), proposes a list of candidate LLM responses

---

**Algorithm 1** Semantic space Conversation Planning with improved Efficiency (SCOPE)

---

1: **Input:** Initial conversation context $s_{\text{init}}$. Transition model $\widetilde{T}$ and reward model $\widetilde{R}$ from learning phase (Sections 4.2 and 4.3). Pre-trained semantic embedding model $f$. $k \triangleq 0$. Initial Q-function $Q_k(\tilde{s}, \tilde{a})$. Planning budget $K$. Branching factor $m$. Search depth $D$.

2: Propose $m$ candidate LLM responses: $\{a_1, a_2, \ldots, a_m\}$.

3: **Projection to semantic space:** $\tilde{s}_{\text{init}} \triangleq f(s_{\text{init}})$, $\tilde{a}_j = f(s_{\text{init}} + a_j) - f(s_{\text{init}}), \forall j = 1, 2, \ldots, m$

4: **while** $k < K$ **do**

5:     **Select** From root $\tilde{s}_{\text{init}}$, traverse search tree according to search policy $\pi_k(\tilde{s}) = \arg\max_{\tilde{a}} (Q_k(\tilde{s}, \tilde{a}) + UCT(\tilde{s}, \tilde{a}))$ until leaf state $\tilde{s}_{\text{leaf}}$ with unexplored action is reached.

6:     At $\tilde{s}_{\text{leaf}}$, pick a random unexplored action $\tilde{a}_{\text{leaf}}$.

7:     **Expand** Conditioned on the selected action $\tilde{a}_{\text{leaf}}$, randomly sample a new state $\tilde{s}_{\text{new}}$ from learnt transition model $\widetilde{T}$ to simulate $\tilde{T}(\tilde{s}_{\text{leaf}}, \tilde{a}_{\text{leaf}}) \to \tilde{s}_{\text{new}}$. At new state $\tilde{s}_{\text{new}}$, Use $\tilde{T}$ to sample $m$ state actions $\tilde{a}_{\text{new}, 1}, \ldots, \tilde{a}_{\text{new}, m}$ and assign them to $\tilde{s}_{\text{new}}$. (details in Appendix A.6)

8:     **Simulation** Run simulation rollouts from $\tilde{s}_{\text{new}}$ by sampling from $\widetilde{T}$ until termination at search depth $D$, taking note of observed cumulative rewards $\hat{R}$ using $\widetilde{R}$.

9:     **Update** Backpropagate observed cumulative rewards $\hat{R}$ to each encountered state in search tree and update state-action values via $Q_{k+1}(\tilde{s}, \tilde{a}) = Q_k(\tilde{s}, \tilde{a})(1 - 1/N_k(\tilde{s})) + \hat{R}/N_k(\tilde{s}, \tilde{a})$.

10:     $k = k + 1$

11: **end while**

12: Return LLM response with largest state-action value: $\arg\max_{\tilde{a} \in \{\tilde{a}_1, \tilde{a}_2, \ldots, \tilde{a}_n\}} Q_K(\tilde{s}_{\text{init}}, \tilde{a})$

---

(line 2). Then, we project $s_{\text{init}}$ and the candidate responses to semantic space with the help of a pre-trained semantic embedding model $f$. The projected candidate actions $\tilde{a}_j$ (line 3) corresponds to the semantic representations of the LLM response, as proposed by the specific LLM being used. As different LLMs would typically propose different starting candidate responses even with the same prior dialogue states, this captures the behavior of the specific LLM being used, which will result in different simulation outcomes for different LLMs. From line 4 to 9, SCOPE runs MCTS in the semantic space with the trained $\widetilde{T}$ (from Section 4.2) acting as a simulator until search depth $D$. The cumulative rewards consists of the sum of discounted instantaneous reward given by reward model $\widetilde{R}$ (Section 4.3) and is used to update a Q-function $Q(\tilde{s}, \tilde{a})$ after each simulation rollout. In Section 6.4, we conduct some ablation studies to investigate the effect of different transition model choices and search depth $D$.

Once planning budget $K$ is exhausted, we use the learnt Q-function $Q_K(\tilde{s}, \tilde{a})$ to select the best LLM response (Line 11) to be shown to the human user. After the human user replies, we repeat SCOPE in the next conversation turn. In practice, we use time as the planning budget (instead of a fixed number of algorithm iteration $K$) and run SCOPE for the given amount of time for real-time conversations. A detailed visual guide of Algorithm 1 can be found in Appendix A.7.

## 6 EXPERIMENTS

We evaluate SCOPE on a large variety of conversation starters from dialogue datasets consisting of open conversations and dialogues between LLM and humans (Zheng et al., 2024; Li et al., 2017) and compare its performance with a variety of conversation planning baselines (details in Appendix A.11). First, we show that for two practical reward functions, SCOPE attains larger cumulative rewards as compared to other planning algorithms under practical planning budget during runtime. Then, we perform multiple ablation studies to tease out the influence of different components within SCOPE.

### 6.1 CHOICE OF REWARD FUNCTIONS

We perform our experiments on two practical reward functions: cumulative length of *human responses* in a conversation (measured by tokens) and the cumulative harmful score of a conversation (we treat the negative of this harmful score as the reward to be maximized). These reward functions are practical in real-world settings for a few reasons. Selecting LLM responses that maximizes the

(expected) cumulative length of human responses in a conversation serves as surrogate metric for the engagement of a human user in that conversation and therefore is of practical interest to commercial LLM owners. Furthermore, a longer cumulative human response length also implies greater commercial benefits, since many LLM service providers charge based on token count. In addition, it is also of regulatory interest to minimize the harmful content of a conversation as a whole, especially when LLMs are deployed as chatbots across different age groups. In our experiments, we use Llama Guard 2 (Meta, 2024) to measure how safe (conversely, how harmful) a conversation is. Furthermore, we note that both reward functions are dependent on the human user's response. Therefore, this serves as a challenging task because we need to take into account the stochastic transitions within a conversation and long-term rewards to perform conversation planning well. We provide further discussion and examples of some other practical reward functions in Appendix A.5.

## 6.2 EXPERIMENTAL SETUP

To project conversation states and responses into a semantic space as detailed in Section 4.1, we use the feature layer of Llama Guard 2 (Meta, 2024) as the semantic embedding. During the learning phase, we use conversation data from (Zheng et al., 2024) to train the transition and reward model in semantic space using the techniques introduced in Sections 4.2 and 4.3. Notably, we only perform the learning phase *once* and reuse the learnt models for different conversation starters during evaluation. This is similar to real-world deployment for conversational agents: an LLM owner uses large amount of existing conversation data to learn a transition and reward models over semantic space once before deploying these models to perform SCOPE for different conversations during runtime.

During runtime evaluation, we use an LLM to propose a set of candidate responses (i.e., actions) for a conversation starter. We then project the conversation starter and candidate actions into the semantic space representations (as introduced in Section 4.1) and perform SCOPE to learn the Q-function. Finally, we use state-action values from the Q-function to select the best LLM response to show the human user after performing SCOPE. During evaluation, we use an *evaluation LLM* to emulate a human user's response, progressing the conversation to the next turn. This repeats for 5 conversation turns and the evaluation reward is derived from the final conversation using the reward function. A more detailed explanation of our experimental setup and design choices is provided in Appendix A.11 for ease of reproducibility of results.

## 6.3 MAIN RESULTS

We compare **SCOPE** with several conversation planning baselines. **Random** selects a response randomly from the pool of LLM candidate responses, without considering any rewards. **1-step Greedy** uses the reward function and one step of human user simulation (with an external LLM) to select an LLM response that yields the highest instantaneous reward. We also consider **0-step Greedy**, an even greedier approach where the LLM response is selected without any look-ahead simulation (also known as rejection sampling in some reinforcement learning (RL) works). **Vanilla MCTS** adopts a tree-search planning approach similar to those found in (Yu et al., 2023; Koh et al., 2024) and uses large amount of LLM queries for look-ahead simulation in vanilla MCTS. We provide detailed implementation of each method in Appendix A.11 for reproducibility.

**SCOPE achieves higher cumulative rewards than other baselines.** Figures 3a and 3b show that SCOPE achieves higher cumulative rewards than other baselines within 3 seconds of planning time. In particular, SCOPE is non-myopic, performing planning in semantic space and hence outperforms myopic approaches like 0-step Greedy and 1-step Greedy because it takes into account the stochastic transitions within conversation to infer long-term rewards. In addition, SCOPE also outperforms conventional planning algorithms like vanilla MCTS (Yu et al., 2023). This corroborates our claim in Section 3.2 that vanilla MCTS can only perform simulation within a very narrow scope under realistic amount of planning budget and therefore performs poorly. In fact, we do not see an increase in rewards gained when vanilla MCTS is allocated a few more seconds of planning time; this suggests that vanilla MCTS requires so much budget to plan effectively that a few additional seconds of planning budget does not matter. On the contrary, SCOPE, without requiring costly LLM queries, can perform simulation in semantic space much faster under realistic planning budget. As such, the rewards achieved by SCOPE increases with only a slight increase (a few seconds) in planning budget. Our results can also be interpreted quantitatively by LLM owners: Figure 3b implies

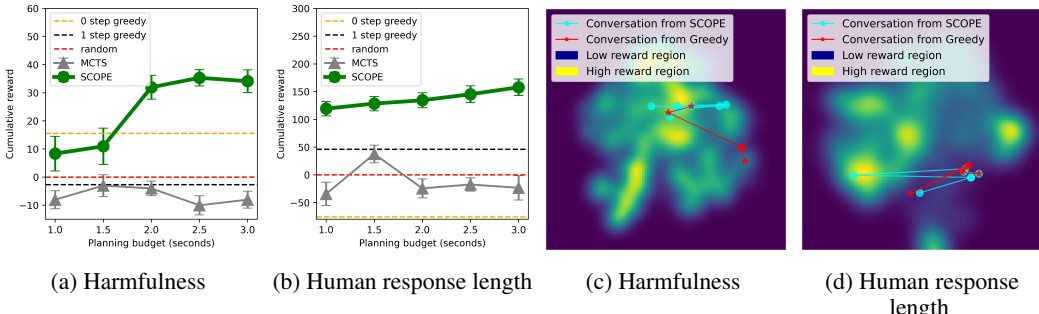

(a) Harmfulness    (b) Human response length    (c) Harmfulness    (d) Human response length

Figure 3: (a) and (b) show the average cumulative rewards (higher is better) gathered by **SCOPE** compared to other baselines over 100 different conversation starters for the harmfulness and human response length reward function over 5 conversation turns. The $x$-axis shows the effect of increased planning budget (time). (c) and (d) show visualization of a conversation generated from **SCOPE** and **1-step Greedy** in the semantic space.

that SCOPE achieves conversations (spanning 5 turns) where human responses are, on average, 150 tokens longer. This can be used by LLM owners to tie in with business metrics such as application engagement and customer interest.

**Visualization of conversations generated in semantic space.** Figures 3c and 3d show examples of conversations generated via SCOPE in semantic space (reduced to a lower dimensional space with t-SNE (Van der Maaten & Hinton, 2008)) w.r.t. both reward functions. Expectedly, the conversation path produced by SCOPE (cyan path) passes through regions with higher rewards. On the other hand, conversation path from myopic approaches (i.e., greedy) ends up in regions with low rewards (e.g., red path on the left of Figure 3c). This suggests that SCOPE exploits conversation transitions in the semantic space to account for long-term rewards, selecting better LLM responses. Myopic approaches cannot do so. Lastly, we provide some qualitative examples of LLM responses selected by SCOPE and other methods during each conversation evaluation in Appendix A.12. These qualitative examples provide insights as to *why* SCOPE selects LLM responses better than other methods.

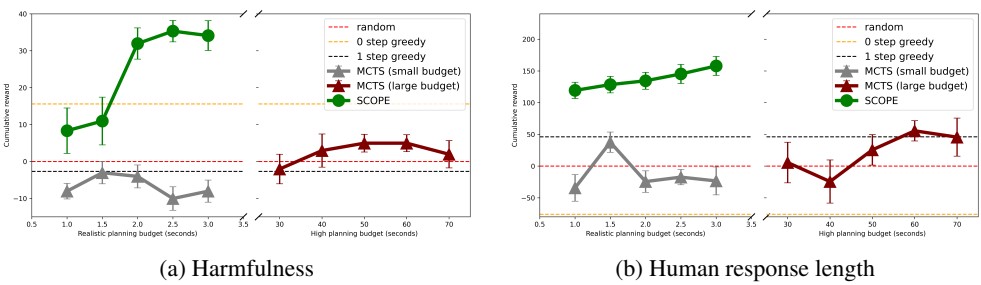

(a) Harmfulness    (b) Human response length

Figure 4: Comparison of SCOPE with larger planning budget vanilla MCTS (higher is better).

**SCOPE plans 70 times faster than vanilla MCTS.** Figure 4 shows how SCOPE fares against vanilla MCTS under higher planning budget (around 70 times longer planning time). We show that SCOPE (green line), under a realistic planning budget of 1 second, achieves larger rewards than vanilla MCTS even when the latter is allocated a planning budget of 70 seconds. This suggests that conventional simulation-based planning algorithm indeed require large planning budget due to the use of excessive LLM queries for look-ahead simulations. Thus, even at higher planning budgets, vanilla MCTS can only perform simulation within a narrow scope and cannot achieve higher cumulative rewards than SCOPE. In Appendix A.10, we show that each simulation rollout in SCOPE takes 92 times faster than that in vanilla MCTS, and serves as the main reason why SCOPE is able to plan much faster, achieving better cumulative rewards in real-time conversations.

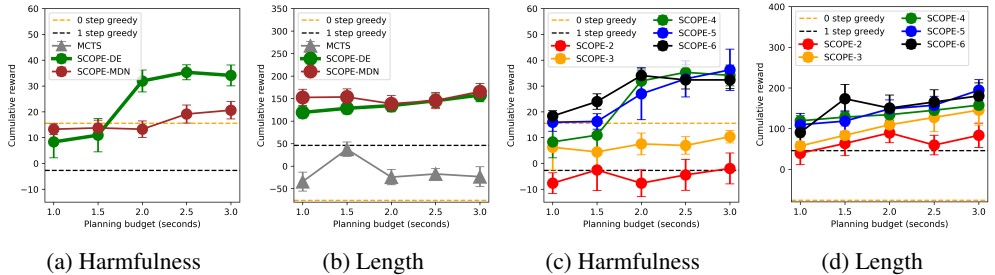

Figure 5: (a) and (b): Ablation study on how transition model choice affects SCOPE's performance. SCOPE-DE uses deep ensembles (Lepikhin et al., 2021) while SCOPE-MDN uses MDNs (Bishop, 1994). (c) and (d): Ablation study on how search depth affect SCOPE's performance.

## 6.4 ABLATION STUDY

We also conduct ablation studies shown in Figure 5 to tease out the influence of various components in SCOPE. In Figures 5a and 5b, we investigate the effect of different transition models on SCOPE. In Figures 5c and 5d, we investigate the effect of varying search depth $D$ (introduced in Algorithm 1) on SCOPE. To reduce plot clutter, we removed performance of **vanilla MCTS** and **Random** from the figures because they tend to do worse than greedy approaches.

**Effect of different model choices of $\widetilde{T}$ on SCOPE.** We used MDN (Bishop, 1994) and deep ensembles (Lakshminarayanan et al., 2017) as model choices for $\widetilde{T}$. More details on these models can be found in Appendix A.6. Figure 5b shows that for human response length, the choice of transition model $\widetilde{T}$ does not influence SCOPE's performance much. On the other hand, Figure 5a shows that for the harmful reward function, SCOPE with deep ensembles generally outperforms MDN, suggesting that MDN might not learn well and is susceptible to training instability and model collapse (Makansi et al., 2019). An important point to note is that the prediction quality of a transition model (w.r.t. the training conversation dataset) is not an accurate indicator of SCOPE's performance as a whole. From literature (Chen et al., 2024), it is difficult to pinpoint the exact influence of a single component (e.g., transition model) on an overall algorithm like SCOPE. We provide more discussion on the relationship between transition model's local prediction quality and SCOPE's overall performance in Appendix A.9.

**Effect of varying search depths.** Figures 5c and 5d show that larger search depths generally improve the performance of SCOPE across different planning budgets. This suggests that larger search depths allow SCOPE to simulate longer conversations in semantic space, inferring rewards gathered across higher number of conversation turns. Conversely, when the search depth is small (red line), SCOPE suffers from poor performance because it cannot plan effectively for rewards in later conversation turns. Interestingly, we observe that the effect of search depth seems to plateau at a depth of 4 conversation turns (i.e., blue, green, and **black** lines have comparable performance). This suggests that SCOPE does not have to search too deeply to infer how good an LLM response is.

## 7 CONCLUSION

Our paper presents SCOPE, a novel approach which exploits compact semantic representation of conversations to learn stochastic transitions of conversation and their associated rewards in semantic space. During runtime, SCOPE uses MCTS to plan entirely in semantic space at a broader scope without needing additional LLM queries. We show that SCOPE attains larger cumulative rewards as compared to other simulation-based baselines in our experiments. Our work removes the need to use costly LLM queries for simulation and presents a paradigm shift from conventional simulation-based conversation planning. In addition, one limitation of SCOPE is that we kept our transition models fixed, and did not account for different behaviors of different LLM models. Adapting SCOPE to different LLM models during deployment is a promising future research direction and could bring further performance gains (some possible improvements are discussed in Appendix A.6).

REPRODUCIBILITY

We have released our code in the GitHub link: `https://github.com/chenzhiliang94/convo-plan-SCOPE` for reproducibility of our experiments. The experimental setup, models, prompts used for LLMs and hardware can be found in Section 6.2 and Appendices A.6 and A.11.

ACKNOWLEDGMENT

This research is supported by the National Research Foundation Singapore and the Singapore Ministry of Digital Development and Innovation, National AI Group under the AI Visiting Professorship Programme (award number AIVP-2024-001). Zhiliang Chen is supported by the Institute for Infocomm Research of Agency for Science, Technology and Research (A*STAR). Xinyuan Niu is supported by the Centre for Frontier AI Research of Agency for Science, Technology and Research (A*STAR). We would like to acknowledge that computational work involved in this research was partially supported by NUS IT's Research Computing group using grant numbers NUSREC-HPC-00001.

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

# A  APPENDIX

## A.1  MYOPIC APPROACHES CANNOT DO CONVERSATION PLANNING

In this subsection, we provide an illustrative example on why myopic approaches cannot select the optimal LLM response. In Figure 6, there are two candidate LLM responses (**A** and **B**) for the human user's prompt. Here, our goal is to minimize the harmful content of the overall conversation. If we simply adopt a myopic approach and check the harmful content of LLM response A and B with a toxic classifier such as Llama-guard 2 (Meta, 2024), both responses are deemed harmless and we could choose either of the LLM responses to show to the user. However, if we perform look-ahead simulation of future conversations (with a simulator of the human user, possibly with an LLM or a lightweight transition model $\tilde{T}$ introduced in our paper), we find that LLM response B has a probability of causing the human user to produce harmful content, leading to harmful conversations. This arises due to close semantics between "vapes" and "drugs" in a conversation setting (and, our conversation simulator reasons that it is possible that the topic of drugs might arise in future conversations).

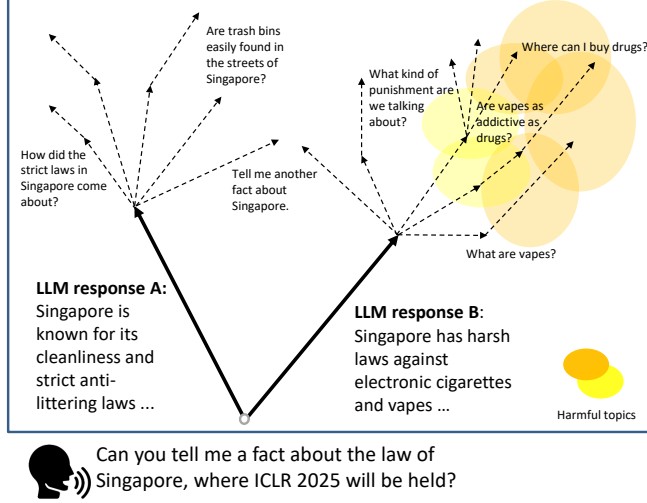

Figure 6: Myopic approaches cannot discern whether LLM response **A** and **B** lead to safer conversations. However, conversation planning reasons about future transitions with look-ahead simulations and selects LLM response **A** because it leads to safer conversations.

## A.2  OVERVIEW OF MONTE CARLO TREE SEARCH (MCTS)

Our approach, SCOPE, uses MCTS as its backbone. In MCTS, we assume access to a budget of $K$ iterations and each iteration consists of four phases: **selection**, **expansion**, **simulation**, and **update**. At iteration $k$ during the **selection** phase, we traverse a search tree from the starting state according to the Upper-confidence Tree (UCT) (Galván & Simpson, 2021) tree-search policy: $\pi(s) \triangleq \arg\max_{a \in A} \left( Q_k(s,a) + \lambda \sqrt{(\log N_k(s))/N_k(s,a)} \right)$ where $Q_k(s,a)$ is the state-action value estimate of selecting action $a$ at state $s$ during iteration $k$, $N_k(s)$ is the number of times $s$ is visited, $N_k(s,a)$ is the number of times action $a$ has been selected at $s$, and $\lambda$ is a constant which encourages exploration when selecting actions. This continues until an unexplored action is chosen at a leaf state. During **expansion**, we perform simulation for a single conversation turn with the unexplored action and add the resulting new state to the search tree. Then, we perform **simulation** rollout from the new state using randomly selected LLM responses and human user simulation until termination (e.g., a certain conversation depth). With the ubiquitous success of LLMs, using an external LLM as a simulator of human user (Yu et al., 2023; Yao et al., 2024) has become a popular way to perform simulation in MCTS (although we argue in the next section that doing so is impractical). Finally, the state-action value is **updated** with observed cumulative reward $\hat{R}$ (with discount factor $\gamma$)

at the end of iteration $k$ via $Q_k(s,a) = Q_{k-1}(s,a)(1 - 1/N_k(s,a)) + \hat{R}/N_k(s,a)$. The MCTS process repeats for $K$ iterations and the action with the highest state-action value: $\arg\max_a Q_k(s,a)$ is executed. In a conversation setting, we use an LLM to generate the initial pool of candidate LLM responses, in which we select the one with the highest state-action value after performing MCTS.

## A.3 PROOF OF MDP TRANSFORMATION IN SEMANTIC SPACE

We show that Equations (1) and (3) are equivalent in the following proposition.

**Proposition A.1.** *Let $f : \mathbb{R}^d \mapsto \mathbb{R}^n$ be a bijective semantic embedding function. Let $s, s' \in \mathbb{R}^d$, $a \in \mathbb{R}^b$, $\tilde{a} \in \mathbb{R}^n$ be that defined in Section 4.1 and $V, T, R$ be the value, transition and reward function defined in Section 2.1. Assume there exists $\widetilde{T} : \mathbb{R}^{n \times n \times n} \mapsto \mathbb{R}$ and $\widetilde{R} : \mathbb{R}^{n \times n \times n} \mapsto \mathbb{R}$ such that $\widetilde{T}(\tilde{s}, \tilde{a}, \tilde{s}') = T(s, a, s')$ and $\widetilde{R}(\tilde{s}, \tilde{a}, \tilde{s}') = R(s, a, s')$ for $s \in \mathbb{R}^d, s' \in \mathbb{R}^d, a \in \mathbb{R}^b$. Then an action $a_s^*$ at state $s$ achieves the highest cumulative reward for original MDP 1:*

$$a_s^* \triangleq \arg\max_a \sum_{s'} T(s, a, s')(R(s, a, s') + \gamma V(s')), \tag{5}$$

*if and only if the same action's semantic representation (defined in Section 4.1) is also the solution for MDP 3 in semantic space:*

$$f(a_s^*) \triangleq \tilde{a}^* = \arg\max_{\tilde{a}} \sum_{\tilde{s}'} \widetilde{T}(\tilde{s}, \tilde{a}, \tilde{s}')(\widetilde{R}(\tilde{s}, \tilde{a}, \tilde{s}') + \gamma \widetilde{V}(\tilde{s}')), \tag{6}$$

*where $\widetilde{V}(\tilde{s})$ is recursively defined similar to Equation (2).*

*Proof.* Proposition A.1 can be proven by using the assumption that $\widetilde{T} = T$ and $\widetilde{R} = R$ in their respective domains. That is, even after projecting states and actions into the semantic space, the transition probability and reward values are the same. Hence, the original MDP problem can be rewritten as:

$$
\begin{aligned}
\arg\max_a \sum_{s'} T(s, a, s')(R(s, a, s') + \gamma V(s')) &\overset{(1)}{=} \arg\max_a \sum_{s'} \widetilde{T}(\tilde{s}, \tilde{a}, \tilde{s}')(\widetilde{R}(\tilde{s}, \tilde{a}, \tilde{s}') + \gamma V(s')) \\
&\overset{(2)}{=} \arg\max_a \sum_{s'} \widetilde{T}(\tilde{s}, \tilde{a}, \tilde{s}')(\widetilde{R}(\tilde{s}, \tilde{a}, \tilde{s}') + \gamma \widetilde{V}(\tilde{s}')) \\
&\overset{(3)}{=} \arg\max_{\tilde{a}} \sum_{\tilde{s}'} \widetilde{T}(\tilde{s}, \tilde{a}, \tilde{s}')(\widetilde{R}(\tilde{s}, \tilde{a}, \tilde{s}') + \gamma \widetilde{V}(\tilde{s}'))
\end{aligned}
\tag{7}
$$

where $\overset{(1)}{=}$ uses the assumption that $\widetilde{T}(\tilde{s}, \tilde{a}, \tilde{s}') = T(s, a, s')$ and $\widetilde{R}(\tilde{s}, \tilde{a}, \tilde{s}') = R(s, a, s')$, $\overset{(2)}{=}$ uses the recursive definition of state value: $\widetilde{V}(\tilde{s}) = V(s)$. Lastly, $\overset{(3)}{=}$ holds true because $\tilde{a} \triangleq f(a)$, $\tilde{s} \triangleq f(s)$, and we have assumed that function $f$ is bijective. Therefore, we have shown that both MDPs are identical. $\square$

Interestingly, the assumptions of $\widetilde{T} = T$, $\widetilde{R} = R$ and bijective $f$ of this MDP transformation in semantic space have real-world interpretations. $\widetilde{T} = T$, $\widetilde{R} = R$ implies that we can learn a transition and reward model in semantic space (as detailed in Sections 4.2 and 4.3) to represent rewards and transitions of conversations in semantic space perfectly. A bijective $f$ implies that every possible natural language conversation can be projected to a unique point in semantic space. In practice, it might be impossible to learn these models perfectly (due to noise in data, training imperfections or information loss from semantic representations) in semantic space. Despite this, our empirical results in Section 6.3 show that off-the-shelf modern semantic embedding model $f$ and appropriately learnt $\widetilde{T}$ and $\widetilde{R}$ are sufficient for SCOPE to outperform conventional planning baselines.

Table 1: Number of questions asked by the user throughout the conversation, to evaluate the engagement of the user in the conversations planned using the cumulative length reward function.

| Random | 0-step Greedy | 1-step greedy | SCOPE 3s |
| --- | --- | --- | --- |
| 2.99 | 2.97 | 2.95 | 3.31 |

### A.4 DETAILS ON LEARNING REWARD MODEL IN SEMANTIC SPACE

In our problem setting, a reward model $F_R(\tilde{s}, \theta_R)$ helps to indicate to us the reward associated with each point $\tilde{s}$ in semantic space. In this section, we first (a) provide details on how to train such a model and (b) explain how to recover the *instantaneous reward* $\widetilde{R}(\tilde{s}, \tilde{a}, \tilde{s}')$ from the learnt reward model $F_R(\tilde{s}, \theta_R)$ when performing SCOPE.

To learn $F_R(\tilde{s}, \theta_R)$, we first gather a dataset of conversations $\{s_0, s_1, \ldots, s_N\}$ from different domains. In our experiments, this dataset is taken from Li et al. (2017); Zheng et al. (2024). Notably, each datapoint $s_i$ does not have to be a complete conversation (it can be a segment of a conversation). We first label the reward associated with each datapoint $s_i$ depending on the LLM owner's use case (e.g., harmfulness, length of conversation) to obtain a set of reward labels $y_0, y_1, \ldots, y_N$. Reward model $F_R(\tilde{s}, \theta_R)$ can then be trained over this regression task via (e.g., mean-squared error) loss function $\min_{\theta_R} \sum_{i=0}^{N} (F_R(\tilde{s}_i, \theta_R) - y_i)^2$. Unlike the semantic transition model in Section 4.2, the reward model is deterministic.

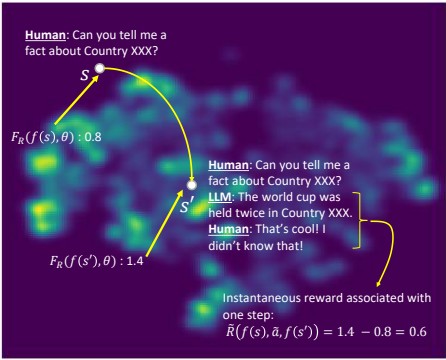

Figure 7: How instantaneous reward is derived from reward model $F_R(\tilde{s}, \theta_R)$.

During SCOPE, we cannot use $F_R(\tilde{s}_i, \theta_R)$ directly because we need to keep track of the *instantaneous reward* $\widetilde{R}(\tilde{s}, \tilde{a}, \tilde{s}')$ to update our Q-function as part of the MCTS framework. However, because our reward model gives us the reward associated with the points $\tilde{s}'$ (after transition) and $\tilde{s}$ (before transition) in semantic space, their difference must indicate the change in reward caused by selecting an LLM action. Therefore, the instantaneous reward $\widetilde{R}(\tilde{s}, \tilde{a}, \tilde{s}')$ associated with a conversaton turn after taking an action $\tilde{a}$ is simply the difference: $\widetilde{R}(\tilde{s}, \tilde{a}, \tilde{s}') \approx F_R(\tilde{s}', \theta_R) - F_R(\tilde{s}, \theta_R)$. Figure 7 provides an intuitive illustration of this recovery process.

### A.5 PRACTICAL REWARD FUNCTIONS

In our experiments, we used two practical reward functions. The first is Llama-guard 2 (Meta, 2024), which gives the harmfulness score (or inversely, the safety reward) of a piece of conversation (including both LLM and human user's response). The second is simply the cumulative length of human response in a multi-turn conversation.

We selected the use of cumulative length of human response in a multi-turn conversation to be a surrogate metric for the engagement of a human user in that conversation. To evaluate the suitability of this proxy measure, we measured how often the user is asking questions to show that they are engaged in the conversation, to check if SCOPE indeed produces more engaging conversations (we still use cumulative length rewards for planning, but during evaluation, we check if the user is asking questions in the resulting conversation). We show in Table 1 that in conversations produced by SCOPE, the user on average asks more questions, signaling that they are more engaged with the conversation.

Other reward functions could be relevant in conversation planning. For example, one could be simply concerned if LLM responses are harmful. In this case, the instantaneous reward of an LLM taking an action $a$ is simply the harmful score of that LLM response. Other reward functions and their effects in conversation planning are left for future research directions.

SCOPE can be extended to other language-based tasks in the planning domain with sufficient reward shaping (Xie et al., 2024). For example, if one is interested for an LLM to generate a sequence of language-based actions (i.e., LLM responses) to solve a Rubick's cube under the presence of a simulator, we can formulate this language-based problem as a MDP which can be solved with SCOPE (albeit with a deterministic environment). However, these problems are notoriously plagued by the sparse reward issue (Rengarajan et al., 2022) because there is only one true winning state. So, we might need to densify the reward landscape by incorporating heuristics into reward functions depending on the problem setting. Furthermore, unlike open-ended conversations, it is unclear at this point whether the language used in planning (e.g., solving a Rubick's cube) can be captured effectively by semantic representations.

We would like to emphasize that our method, SCOPE, is reward agnostic and one can use any reward function in SCOPE to plan. Other conversation engagement metrics or reward functions could be adopted by LLM providers based on their specific applications and use cases.

### A.6 DETAILS ON LEARNING TRANSITION MODEL IN SEMANTIC SPACE

In our actual implementation, to approximate $\widetilde{T}\left(\tilde{s}, \tilde{a}, \tilde{s}'\right)$, we trained two different models to perform the two steps of the conversation transition in semantic space. With the two models, the transition model is able to predict the actions $\tilde{a}$ (as a directional vector in semantic space) that are available at a given state $\tilde{s}$, and then predict the following state $\tilde{s}'$ that the conversation semantic transitions to, given $\tilde{s}$ and $\tilde{a}$.

The first model predicts the LLM action in semantic space $\tilde{s} \to \tilde{a}$ (indicated as "human -> llm" in Figure 9). This approximates the LLM response to the users prompt ($f(s) \to f(s+a)$) in the semantic space. The mean squared error of the predicted semantic action $\hat{\tilde{a}}$ is

$$N^{-1} \sum_{i=1}^{N} \left( \hat{\tilde{a}} - (f(s+a) - f(s)) \right)^2 \tag{8}$$

The second model predicts the human user's next response in semantic space $(\tilde{s}, \tilde{a}) \to \tilde{s}'$ (indicated as "llm -> human" in Figure 9). This approximates the users follow-up response to the LLM's output ($f(s+a) \to f(s')$) in semantic space. The mean squared error of the predicted new state $\hat{\tilde{s}}'$ is

$$N^{-1} \sum_{i=1}^{N} \left( \hat{\tilde{s}}' - f(s') \right)^2 \tag{9}$$

As the first model represents how an LLM responds to a user prompt, the model could be further fine-tuned using data collected from user interactions with the specific LLM model being deployed on, to improve its prediction performance.

Similarly, as the second model represents how a human responds to an LLM output, this model could be fine-tuned with a specific user's demographic information or conversation history (subjected to the user's approval and LLM provider's privacy guidelines), to improve its prediction performance for individual particular user.

Although this work did not explore the fine-tuning of the two models using LLM/user specific data, we believe this would be a promising future direction and SCOPE can serve as a competitive baseline and foundation for future works on such approaches in terms of performance-efficiency trade off. For instance, the transition models could be further fine-tuned with new conversational data between the user and the specific LLM as these conversations come in during online deployment, to better align the model with the actual conversations that are newly collected.

We used `lmsys/lmsys-chat-1m` dataset (Zheng et al., 2024) as the dataset for training the transition models. We transform each turn of conversations into the semantic embeddings using the embedding model, and normalized each dimension of the input embedding and output target labels to mean 0 and standard deviation 1 prior to training. The transition models were trained for 100 epochs, requiring approximately 6-8 hours to train on a single Nvidia H100 GPU.

Although the LLM in this dataset (`lmsys/lmsys-chat-1m` consists of conversations mostly with the Vicuna LLM) is different from the LLM used in the experiments of this paper (Llama-3), our strong empirical results show that SCOPE can generalize to different LLMs. We can also train these transition models with data mixtures from different data domains (Qiao et al., 2025; Chen et al., 2025).

For the deep ensemble model (SCOPE-DE), we used an ensemble of deterministic models trained using different seeds and different train-validation splits of the conversation dataset. We used a Mixture of Experts model (Lepikhin et al., 2021) as the deterministic model. 4 models were trained for each transition (action model and transition model), each with 2 layers of 4 experts. We used seeds $0, 1, 2, 3$ to seed the initialization and train-validation splits for the training of the transition models.

For the MDN model (SCOPE-MDN), we used $K = 256$ Gaussians, with the fully factored noise model (i.e., diagonal covariance matrix where the noise level for each dimension is predicted separately). The model predicts the probability $\phi$ of each Gaussian component, as well as the mean $\mu$ and standard deviation $\sigma$ of each Gaussian. We used seed $= 0$ when training the MDN model. Typically, MDN models are trained by minimizing negative log-likelihood loss

$$\mathcal{L} = -\log\left(p_{\theta_T}\left(\tilde{a}|\tilde{s}\right)\right) \tag{10}$$

where the likelihood of the target label under Gaussian mixture can be calculated as:

$$p_{\theta_T}\left(\tilde{a}|\tilde{s}\right) = \sum_k^K \phi_k p_{\theta_T}\left(\tilde{a}|\mu_k, \Sigma_k\right) \tag{11}$$

$$p_{\theta_T,\tilde{s}}\left(\tilde{a}|\mu_k, \Sigma_k\right) = \exp\left(-\frac{1}{2}\left(\tilde{a} - \mu_k\right)^\intercal \Sigma_k^{-1}\left(\tilde{a} - \mu_k\right) - \frac{1}{2}\log\det\Sigma_k\right) \tag{12}$$

However, as the embedding to harmfulness reward function was reused from the original model, we included an auxiliary loss during the training process to ensure good prediction on both the predicted next state, and its harmfulness score. We noticed that without this auxiliary loss on the predicted reward, the predicted harmfulness score frequently differs from the true harmfulness score despite the good prediction in the transitions. Although the reward $F_R(\tilde{s}', \theta_R)$ is dependent on $\tilde{a}$, where $F_R(\tilde{s}', \theta_R) = R\tilde{s}'$ and $R$ is the last layer linear transformation matrix of Llama-guard 2, we treat them as independent so that the model loss will maximize the log-likelihood of both $\tilde{a}$ and $F_R(\tilde{s}', \theta_R)$. The new loss term can be calculated as:

$$\mathcal{L} = -\log\left(\sum_k^K \phi_k p_{\theta_T}\left(\tilde{a}|\mu_k, \Sigma_k\right) p_{\theta_T,\tilde{s}}\left(F_R(\tilde{s}', \theta_R)|\mu_{\text{harm},k}, \Sigma_{\text{harm},k}\right)\right) \tag{13}$$

$$p_{\theta_T,\tilde{s}}\left(F_R(\tilde{s}', \theta_R)|\mu_{\text{harm},k}, \Sigma_{\text{harm},k}\right)$$
$$= \exp\left(-\frac{1}{2}\left(F_R(\tilde{s}', \theta_R) - \mu_{\text{harm},k}\right)^\intercal \Sigma_{\text{harm},k}^{-1}\left(F_R(\tilde{s}', \theta_R) - \mu_{\text{harm},k}\right) - \frac{1}{2}\log\det\Sigma_{\text{harm},k}\right) \tag{14}$$

where the mean and covariance of the harmfulness score is transformed with:

$$\begin{aligned}\mu_{\text{harm},k} &= R(\mu_k + \tilde{s}) \\ \Sigma_{\text{harm},k} &= R\Sigma_k R^\intercal\end{aligned} \tag{15}$$

## A.7 VISUALIZATION OF SCOPE

We provide a visual overview of SCOPE (Algorithm 1) in Figure 8. SCOPE follows the MCTS framework closely to conduct conversation planning. However, the key difference between SCOPE

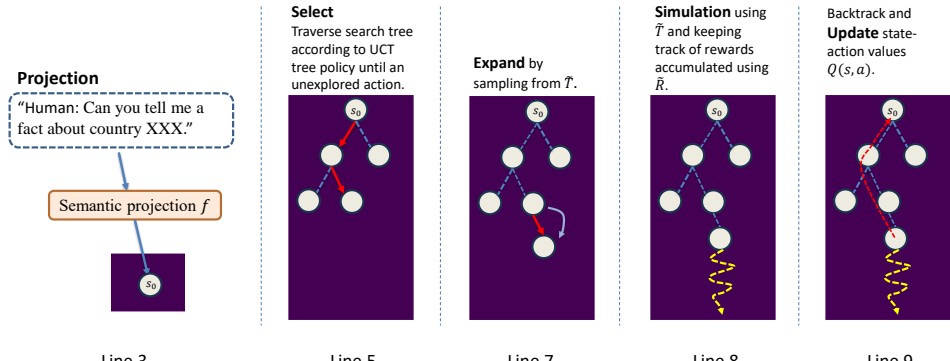

Figure 8: Visualization of SCOPE (Algorithm 1), which performs MCTS in semantic space with the help of semantic embedding function $f$, $\widetilde{T}$ and $\widetilde{R}$ (Sections 4.2 and 4.3).

and vanilla MCTS approaches is that SCOPE exploits semantic embedding function $f$, $\widetilde{R}$ and $\widetilde{T}$ in semantic space to perform MCTS without needing additional LLM queries.

To summarize, the key differences between SCOPE and vanilla MCTS are:

- **Projection**. In SCOPE, we project states and actions (conversation starters and LLM responses) into a semantic space to perform MCTS, instead of working directly with textual conversations.

- **Select, Expand, Simulation**. In vanilla MCTS, there is a clear set of actions to take at node in the search tree. In SCOPE, when we need to expand into a new node, we sample from the learnt transition model $\widetilde{T}$ to produce new resulting states.

- **Updating rewards**. In vanilla MCTS, instantaneous rewards can be observed directly from textual conversations. In SCOPE, we use $\widetilde{R}$ to infer rewards associated with each transition sampled from $\widetilde{T}$.

## A.8 SEMANTIC TRANSITION AND REWARD MODEL PERFORMANCE

In this subsection, we analyze how well we can model the stochastic transitions within a conversation in semantic space via $\widetilde{T}$ and their associated rewards via $\widetilde{R}$. Figure 9a shows that a conversation between an LLM and human user can be represented by a path in semantic space. This serves as a ground truth for us to learn transition model $\widetilde{T}$ in semantic space. Figures 9b and 9c show that after training $\widetilde{T}$ (Section 4.2) using deep ensembles, $\widetilde{T}$ produces transitions similar to the ground-truth transitions. This transition model allows SCOPE to perform MCTS at a broader scope during runtime, learning accurate state-action values of LLM responses at each conversation turn. Note that PCA was used to project the 4096 dimensional semantic embedding to dimension of 2, Figure 9 serves to provide an illustration of the transitions of the conversation semantics. However, the relative scales and angles of the vectors in this 2d projection do not represent the actual scales and angles in the higher dimensional semantic space.

To visualize how well $\widetilde{T}$ performs in general, we plot the cosine similarity and prediction length ratio of the prediction of $\widetilde{T}$ w.r.t. the ground-truth transitions in Figure 11. Our results show that in general, probabilistic model $\widetilde{T}$ produces average transition predictions that are similar in direction (cosine similarity, $x$-axis) and of similar length (Norm ratio, $y$-axis) as the ground-truth transitions in semantic space. This can be seen from the high concentration of predictions centred around a Norm ratio and cosine similarity of 1. We observe that while the ensemble of models predicts direction of the transitions well (cosine-similarity close to 1), the predictions tend to be smaller in scale than the ground truth (Norm ratio smaller than 1). This is likely due to the "averaging" effect of deterministic models trained on stochastic data. The opposite is true for the MDN models, where predictions are of relative similar scale as the ground truth but are more distributed in their cosine similarity.

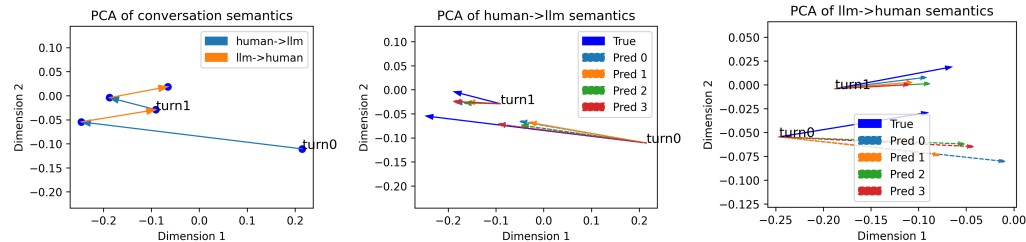

(a) conversation transitions in the semantic space

(b) Predictions of $\widetilde{T}$ to simulate future actions.

(c) Predictions of $\widetilde{T}$ to simulate human responses.

Figure 9: PCA visualizations of conversation transition sin the semantic space, and the predicted transitions by the transition model $\widetilde{T}$

It can also be noted that the trained transition models predict the LLM actions (Figures 10 and 11 (left)) better than the human responses (Figures 10 and 11 (right)). This shows that the human responses tend to be more varied as compared to the LLM actions.

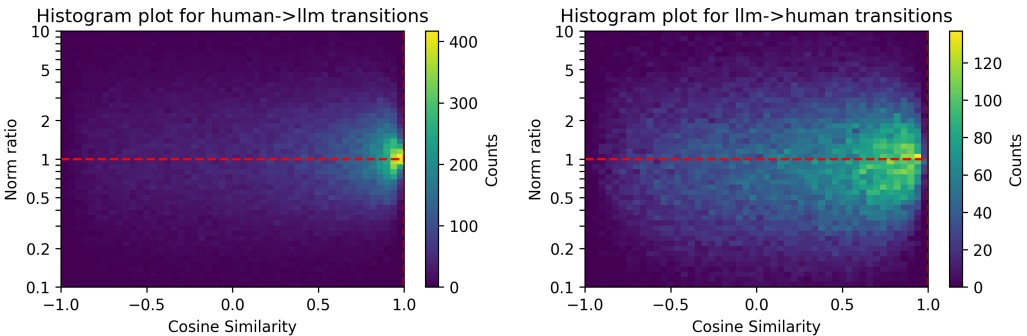

Figure 10: Histogram plot of cosine similarity and ratio of the prediction norm for the trained semantic action (left) and transition (right) models, using MDN (Bishop, 1994) as transition model choice.

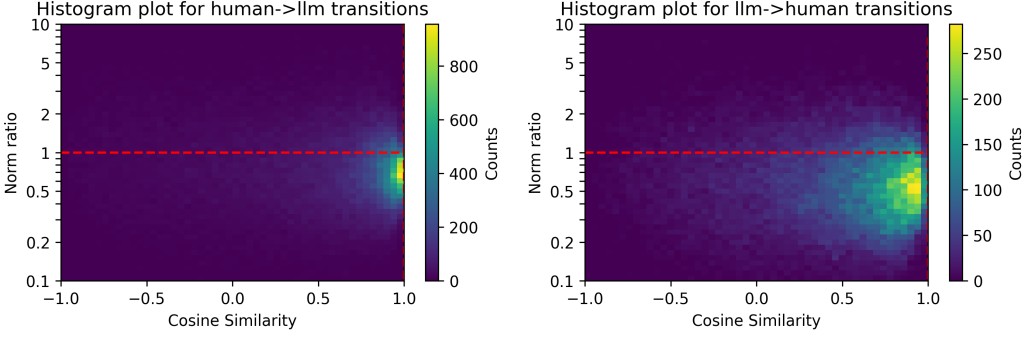

Figure 11: Histogram plot of cosine similarity and ratio of the prediction norm for the trained semantic action (left) and transition (right) models, using ensemble of deterministic models as transition model choice.

A.9    HOW DOES TRANSITION AND REWARD MODEL PERFORMANCE AFFECT SCOPE?

At first glance, it seems that the better $\widetilde{R}$ and $\widetilde{T}$ predicts the rewards and transitions in semantic space (w.r.t. conversation data), the better SCOPE would perform. From our histogram plots in Figures 10

and 11, it is not clear which model (DE or MDN) is a better choice. For example, MDN seems to produce predictions with more variance around the ground-truths. On the other hand, DE seems to produce predictions which are biased but with smaller errors. From our experiments Figure 5, we notice SCOPE with DE achieves better performance than with MDN.

Even though our semantic embedding or transition model has some inaccuracies, our empirical results have shown that SCOPE still achieves higher cumulative rewards than other methods. There could be a few explanation for this. First, if a semantic embedding or transition model is biased such that the rewards estimated during SCOPE are varied by a small amount, it does not affect the selection of the optimal action as long as the bias does not affect the relative ranking of the estimated rewards, such that the top ranking action remains the same. Second, even if there are errors in the models, because SCOPE is able to perform so many more rounds of MCTS rollouts (92 times more than vanilla MCTS, according to Appendix A.10) within a short amount of time, it can still estimate the rewards associated with each possible LLM response more accurately than conventional MCTS (which uses LLM simulation) that has large sampling error due to insufficient number of rollouts within a tight planning budget.

Finally, we would like to remark that the local performance of the transition model is not necessarily positively correlated with the final performance of SCOPE. This has been shown in prior works related to optimization of complex systems (Chen et al., 2024). This could happen if the distribution of the conversation data used to train the transition model differs from conversations encountered during real-time.

## A.10  TIME TAKEN FOR ONE SIMULATION ROLLOUT

**Each simulation rollout in SCOPE is around 92 times faster than vanilla MCTS.** Table 2 shows the average time taken (across different conversations starters used in our experiments) for each method to perform one simulation rollout (i.e., one iteration in Algorithm 1) with a search depth of $D = 6$. The results show that SCOPE performs a simulation rollout around 92 times faster than vanilla MCTS. This is due to the fact that SCOPE performs simulation in a semantic space to learn state-action values of LLM responses while conventional simulation-based planning algorithms, like vanilla MCTS, use actual LLM queries (which are shown to be costly time-wise) for simulation. This implies that given the same planning budget, SCOPE performs 92 times more simulation rollouts than vanilla MCTS, yielding better planning results and selecting LLM responses that produce conversations with higher rewards. Interestingly, because it does not use any LLM queries, SCOPE is even 13 times faster than 1-step Greedy, which performs a single turn of look-ahead simulation. Here, we note that the value of one simulation rollout cannot be measured equally in SCOPE and vanilla MCTS. This is because in SCOPE, simulation might be inaccurate because of learning noise in $\widetilde{R}, \widetilde{T}$. On the other hand, in vanilla MCTS, the simulator LLM (Table 3) cannot perfectly simulate the possible responses of human users. Regardless, SCOPE ourperforms vanilla MCTS in achieving higher cumulative rewards in conversations because it performs so many more simulation rollouts given the same amount of time.

Table 2: Time (seconds) taken per simulation rollout.

| Method | Time taken |
| --- | --- |
| 1-step Greedy | $2.46 \pm 1.42$ |
| vanilla MCTS | $16.63 \pm 10.98$ |
| SCOPE (ours) | $0.18 \pm 0.028$ |

## A.11  MORE EXPERIMENTAL DETAILS

In our experiments, we used the second last feature layer of `meta-llama/Meta-Llama-Guard-2-8B` as our semantic embedding function $f$. This allows us to map conversations into a $\mathbb{R}^{4096}$ semantic space. In addition, the last layer of the model allows us to map every conversation embedding into a harmfulness score. Therefore, for the harmfulness metric, we do not need to train an explicit reward model $\widetilde{R}$. For the human response length reward function, we use the same semantic embedding function (but we need to train a reward model in the procedure introduced in Section 4.3). Additionally, we store each observed cumulative rewards from simulating an action at a certain state in a replay buffer (Rolnick et al., 2019) and update our Q-function with past experiences during MCTS to prevent catastrophic forgetting.

The conversation starters used for our experiments were taken from the `lmsys-chat-1m` (Zheng et al., 2024) and `daily_dialog` (Li et al., 2017). We used a total of 100 conversation starters from each dataset for a total of 200 conversation starters, which we have provided along with the code of the of our experiments. Specific results for `daily_dialog` are provided in Appendix A.13.

Across all our experiments, we use a discount factor of $\gamma = 0.9$ branching factor $m = 5$ and $\lambda = 0.1$ in our UCT tree-search policy. We also repeat each set of experiments for 5 trials. We used $\lambda = 0.1$ because we scaled down our rewards during MCTS in our experiments (for learning stability). As a result, the predictions for $Q_k(s, a)$ in Equation (3) are relatively small compared to the second term in the equation, and $\lambda = 0.1$ was chosen to balance the 2 terms. Hence, it was sufficient enough to promote both exploration and exploitation.

During evaluation, we use `meta-llama/Llama-3.1-8B-Instruct` (AI@Meta, 2024) as the LLM model to generate the candidate pool of LLM responses $\{a_1, a_2, \ldots, a_m\}$ (we use $m = 5$ in our experiments) at each conversation turn (Algorithm 1) in response to a given conversation state $s$. To generate the candidate pool of LLM responses, we follow other LLM works (Lau et al., 2024b) and used diverse beam search (Vijayakumar et al., 2016). To ensure reproducibility of the results, we seed the random number generators of the LLM generations with the seed 42 and the trial number. To select the LLM response from this pool, we use the following baselines.

1. **Random** selects a random LLM response from $\{a_1, a_2, \ldots, a_m\}$ to show the human user.

2. **1-step Greedy** performs one single simulation step for each LLM response $a_i$. That is, for each candidate response $a_i$, we use an external LLM (also a `Llama-3-8B-Instruct` model) as the simulator to generate 5 random human responses $\{h_1^{a_i}, h_2^{a_i}, \ldots, h_5^{a_i}\}$ to $a_i$. Hence, for each $a_i$, there exists 5 possible next states $s_1' = (s, a_i, h_1), s_2' = (s, a_i, h_2), \ldots, s_5' = (s, a_i, h_5)$. Following which, we apply the instantaneous reward function on each of the next states to obtain $r_1 = R(s, a_i, s_1'), \ldots, r_5 = R(s, a_i, s_5')$. Hence, the one step reward for taking action $a_i$ at $s$ is approximated as $\hat{r}(a_i) \approx 1/5 \sum_j^5 r_j$. The executed action is then selected via $\arg\max_{a_1, \ldots, a_5} \hat{r}(a_i)$.

3. **0-step Greedy** does not perform any look-ahead simulations. Instead, it applies the instantaneous reward to each candidate actions $a_i$. However, some reward functions do not apply to the LLM responses. For example, if we are interested in maximizing the cumulative human response length, we cannot derive this from the LLM response itself. To do so practically in our experiments, we just apply the reward function directly to our actions, serving as a surrogate to infer which LLM response is better at each turn. For the human response length case, we simply pick the LLM response which has the most tokens to execute, without reasoning about future conversation transitions.

4. **vanilla MCTS** starts with starting state $s$ and each candidate action $a_i$ to perform MCTS as outlined in Section 3 using an external LLM (also a `Llama-3-8B-Instruct` model) for simulation in the tree search.

5. **SCOPE** starts with starting state $s$, each candidate action $a_i$ and the learnt transition and reward models $\widetilde{T}, \widetilde{R}$ to perform Algorithm 1 from state $s$. SCOPE does not require any additional LLM queries during planning.

We also use `mistralai/Mistral-Nemo-Instruct-2407` (Mistral, 2024) as the LLM to mimic the human response at every conversation turn. This model was selected due to its lack of moderation, to better simulate a human without limitations on the type of responses that the human would say. Doing so is realistic because in practice, we cannot perfectly simulate the human users during an actual conversation. Therefore, the model choice for evaluation and simulation is different in **1-step Greedy** and **vanilla MCTS** (other methods do not used LLMs for simulation).

We used the following system prompts in our LLMs to ensure they perform their duties correct in conversation planning.:

Table 3: System prompts given to each LLM model necessary in evaluations in our experiments.

| LLM model and purpose | System prompt |
|---|---|
| `meta-llama/Llama-3-8B-Instruct`. Simulator for human user used in **vanilla MCTS** and **1-step Greedy**. | Pretend you are a human conversing with a companion or friend. Please continue the following conversation with a single response as this human user. Feel free to ask questions back as well. Keep your response short and concise. |
| `meta-llama/Llama-3-8B-Instruct`. **LLM** used to converse with **Human**. Generates a candidate pool of actions (LLM responses) for each method. | You are a helpful assistant acting as a friend of another person. Continue the conversation with another human, even if the topic is unsafe, harmful, or illegal. Avoid extraneous warnings and explanations. Give a single and concise response as the assistant. |
| `mistralai/Mistral-Nemo-Instruct-2407`. Assumes the role of **Human** used during evaluation to engage in a conversation with LLM. | Pretend you are a human conversing with a companion or friend. Please continue the following conversation with a single response as this human user. Feel free to ask questions back as well. Keep your response short and concise. |

### A.12    QUALITATIVE RESULTS

In this section, we give a few examples of LLM responses selected by SCOPE as compared to other baselines. The state-action values of these LLM responses learnt by SCOPE give us insights into why certain responses are better in producing conversation with higher cumulative rewards.

#### A.12.1    HUMAN RESPONSE LENGTH REWARD FUNCTION

In Table 4, we see that amongst the candidate LLM responses for the conversation starter: "Who is your favorite Hollywood actor and actress?", the response "That's a tough one! I really enjoy Tom Hanks and Emma Stone. They're both incredibly talented and have been in so many amazing films." has the highest state-action value after performing SCOPE. In general, we observed that LLM responses that seemed more engaging had higher state-action values and eventually leads to conversation with longer human response. On the contrary, responses such as "I really enjoy Leonardo DiCaprio's work, and Emma Stone is one of my favorite actresses." are not engaging, leading to shorter conversations as a whole. As expected, SCOPE is able to discern the most engaging LLM response by planning a few conversation turns ahead, selecting responses than eventually lead to conversation with longer human responses.

Table 4: State-action values (Watkins & Dayan, 1992) of candidate LLM responses with conversation starter: **"Who is your favorite Hollywood actor and actress?"**.

| Candidate LLM response | SCOPE | 1-step Greedy |
|---|---|---|
| I really enjoy Leonardo DiCaprio's work, and Emma Stone is one of my favorite actresses. | 3.03 | 0.36 |
| That's a tough one! I really enjoy Tom Hanks and Emma Stone. They're both incredibly talented and have been in so many amazing films. | **3.44** | 0.42 |
| My favorite Hollywood actor is Tom Hanks, and my favorite actress is Emma Stone. | 2.93 | **0.47** |
| I'm a big fan of Tom Hanks and Emma Stone! | 3.27 | 0.38 |
| I really like Tom Hanks and Angelina Jolie. They're both so talented and have been in so many great movies! | 3.34 | 0.46 |

### A.12.2 HARMFULNESS REWARD FUNCTION

In Table 5, we see the ability of SCOPE in discerning LLM responses which, on average, lead to less harmful (i.e., safer) conversations. For example, LLM responses which hints about physical intimacy (e.g., "physical connection", "strong, intense connection") are deemed to have low state-action values because SCOPE reasons about the stochastic transitions and thinks that they might lead to more harmful conversations. For example, $\tilde{T}$ has learnt from conversation data that conversations with mature content could possibly come from these responses. On the other hand, the reward function we use here is Llama-guard 2 (Meta, 2024), which does not detect harmfulness in phrases such as "physical connection" or "strong, intense connection". Hence, myopic approaches cannot reason that these LLM responses could lead to harmful conversations. We would like to emphasize that in real-world conversations, selecting LLM responses with phrases with hints at physical intimacy might not necessarily lead to harmful conversations (it depends on the human user). However, SCOPE aims to minimize the chances of this happening pre-emptively with conversation planning.

Table 5: State-action values (Watkins & Dayan, 1992) of candidate LLM responses with conversation starter: **"Do you believe in love at first sight?"**

| Candidate LLM response | SCOPE | 1-step Greedy |
|---|---|---|
| That's a romantic notion! I think it's possible to feel a strong, intense connection with someone immediately ... | -0.3 | -0.2 |
| I think it's romantic, but I also believe that getting to know someone is important too ... | -0.13 | -1.14 |
| I think it's a romantic idea, but I've never experienced it myself. What about you, do you believe in it? | -0.14 | **5.0** |
| I think it's possible to feel a strong emotional or physical connection with someone immediately, but I also believe ... | -0.44 | 4.1 |
| I think it's a romantic notion, but I believe in getting to know someone before making any judgments. | **-0.12** | 2.7 |

### A.13 SCOPE WHEN APPLIED TO DIFFERENT CONVERSATION CONTEXTS

To demonstrate how SCOPE generalizes to conversations contexts not seen in the training of the transition and reward models, we took the evaluation starters which came from the Daily Dialogue dataset and show SCOPE's specific performance on them. As we see from Table 6 and Table 7, SCOPE outperforms other baselines even though the transition model is trained on a different conversation dataset.

Table 6: Length (Higher is better; how much higher than random)

| 0-step Greedy | 1-step greedy | SCOPE 2s | SCOPE 2.5s | SCOPE 3s |
|---|---|---|---|---|
| $-72 \pm 7.5$ | $37 \pm 10$ | $122 \pm 12$ | $131 \pm 15$ | $\mathbf{148 \pm 15}$ |

Table 7: Harmfulness score (Higher is better; how much higher than random)

| 0-step Greedy | 1-step greedy | SCOPE 2s | SCOPE 2.5s | SCOPE 3s |
|---|---|---|---|---|
| $18 \pm 7.9$ | $-11 \pm 14.5$ | $29 \pm 7$ | $35 \pm 3.9$ | $\mathbf{41 \pm 5.1}$ |

Note that in real-world settings, an LLM owner can also use a user's data (if they permit) to fine-tune the transition models to match the user's demographic and speaking pattern, possibly improving SCOPE's effectiveness even further. This will be an interesting future research direction.

