# OpenReview forum: "Broaden your SCOPE! Efficient Multi-turn Conversation Planning for LLMs with Semantic Space"
_ICLR.cc/2025/Conference — ICLR 2025 Spotlight_

### Official Review · Reviewer_Cm3A · 2024-10-22

**Soundness:** 2
**Presentation:** 3
**Contribution:** 3
**Rating:** 6
**Confidence:** 4

**Summary:**

This paper focuses on speeding up MCTS in semantic space to improve dialogue policy planning. The author proposes SCOPE, a method to convert dialogue into transition/reward functions in semantic space (i.e., embedding space), and then conduct MCTS to determine the optimal response. Specifically, SCOPE obtains the transition function by 1) convert dialogues into embeddings using LLaMA-2 Guard, and then 2) train a model to model state transition using an existing conversation dataset. Then, SCOPE obtains the reward function by similarly training a model to predict the reward associated with each state in the semantic space. Finally, the author evaluated SCOPE against methods such as rejection sampling (i.e., 0-step greedy) and MCTS, and show that SCOPE can achieve superior performance with significantly less compute.

**Strengths:**

- Conducting MCTS in semantic space by modeling the transitions/reward functions in semantic space is novel. As the author mentioned, such an approach significantly reduces the search time from MCTS while retaining a high performance (if the transition/reward can be learned well)

- The authors supported many subtle claims with empirical evidences/theoretical analysis (in Appendix). For example, Appendix A.7 provides additional details to verify the effectively of using probabilistic models for stochastic transitions, and Appendix A.2 presents theoretical justifications for the optimal solution in semantic space, and more. This indicates that the proposed method/problem has been well thought and studied.

- The authors evaluated their approach against popular methods such as MCTS, and showed improvement in performance despite using much less test-time compute.

**Weaknesses:**

1. While the authors argue that "our paper focuses on the training-free setting and explores inference time strategies" (L59), SCOPE is not training free, as it requires training the transition and reward model before test time. This makes direct comparison (e.g., performance v.s. speed) to prompt-based MCTS unfair, as the latter strictly uses no training.

2. This work trains a transition function to predict $T(s) \to (a',s')$ instead of $T(s, a') \to s'$, based on description in L287-293. This means that this transition function needs to *predict both the response that will be generated by the LLM and next the corresponding user response*. This seems unrealistic because 1) if it can accurately model $a'$ then it essentially becomes an LLM, and 2) the planning process becomes *policy agnostic* (also see Algorithm 1 line 7) - a sign indicating that SCOPE may not be robust against using different LLMs as policy models (unlike prompt based MCTS).

3. Since SCOPE requires a trained transition and reward function in latent space, it becomes questionable whether SCOPE can generalize when *evaluation dialogues become OOD compared to the ones used to train the transition/reward function*; or when different LLMs is used to propose candidates at test time.

4. Since SCOPE planning is conducted in latent semantic space, there is a lack of transparency/explanability in its decision making process. This is in contrast to approaches that plans in text space (e.g., prompt based MCTS). This could present difficulties to researchers or users to understand how or why certain actions were chosen.

**Questions:**

Questions:

- In experiments you used $\lambda=0.1$ for UCT, which forces the tree search to focus on exploitation instead of exploration. This is rather an uncommon value. Is there a reason for this?
- Can you provide more details about the benchmarks you tested? Currently its only mentioned in L363-365 as "dialogue datasets consisting of open conversations and dialogue between LLM and humans". Are these generic dialogues from existing chat datasets or are these curated from certain dialogue planning benchmarks?

Comments and Typos:

- Planning in semantic/latent space (L108-111) has been explored in some prior work [1-2]. These should be mentioned in this paper as related work.
- In L259 and L346, it should be "conversation states s" instead of "conversation starter s"
- Currently Introduction and Background/Related work takes up more than 4 pages. This is too long, as it leaves little room for methods and experiments. I would suggest the authors to trim Section 1-3 as much as possible (e.g., details about MCTS can be moved to appendix).
- "Section 6.5 Conclusion" should be "Section 7 Conclusion".
- If I understood correctly, "0-step greedy" directly chooses the best response according to the reward model? If so, this should be named "rejection sampling" instead, which is a common approach used in many RL related work.


---

References

[1] Lubis, Nurul et al. “LAVA: Latent Action Spaces via Variational Auto-encoding for Dialogue Policy Optimization.” ArXiv abs/2011.09378 (2020): n. pag.

[2] Vlastelica, Marin et al. “Taming Continuous Posteriors for Latent Variational Dialogue Policies.” AAAI Conference on Artificial Intelligence (2022).

---

> ### Author Response · Authors · 2024-11-19
>
> ## Part 1/3
>
> We would like to thank the reviewer for the comprehensive review and compliments of our method's novelty and motivation.
>
> > The authors supported many subtle claims with empirical evidences/theoretical analysis (in Appendix). For example, [...] This indicates that the proposed method/problem has been well thought and studied.
>
> First of all, we would like to thank the reviewer for these praises, it means a lot to us as researchers. We will provide additional clarifications to the reviewer's questions in our responses below.
>
> ---
>
> > While the authors argue that "our paper focuses on the training-free setting and explores inference time strategies" (L59), SCOPE is not training free, as it requires training the transition and reward model before test time. This makes direct comparison (e.g., performance v.s. speed) to prompt-based MCTS unfair, as the latter strictly uses no training.
>
> Thank you for the comment. In the context where that sentence was written, we were qualitatively comparing SCOPE with methods such as RLHF, which require us to fine-tune an LLM. That is why we claimed it was "training-free" w.r.t. LLM training. We will improve the writing to make the distinction clear that our method still needs some degree of training (but not on the LLM).
>
> Even though our method needs to train a reward/transition model, we show in our paper that these models are relatively lightweight and can be trained beforehand (we only took an hour to train the transition model, and inference time is negligible) and kept fixed during inference time. In addition, for metrics such as harmfulness by the Llama-guard model, the mapping between semantic space and rewards is taken directly from the pre-trained model's weights and so we do not need to train a separate reward model. Hence, we believe it is still fair to compare the performance/speed of our method with vanilla MCTS. Our aim is to show that we can use a lightweight transition and reward model (both of which are fast to train) with almost no inference overhead to achieve faster and better conversation planning in real time. We will improve the writing of our revised paper by incorporating these clarifications.
>
> ---
>
> > This work trains a transition function to predict $T(s) \rightarrow (a',s')$ instead of $T(s,a') \rightarrow s'$, based on description in L287-293. This means that this transition function needs to predict both the response that will be generated by the LLM and next the corresponding user response. This seems unrealistic because 1) if it can accurately model then it essentially becomes an LLM, and 2) the planning process becomes policy agnostic (also see Algorithm 1 line 7) - a sign indicating that SCOPE may not be robust against using different LLMs as policy models (unlike prompt based MCTS).
>
> Thank you for the question. I think our usage of $T(\tilde{s},\tilde{a},\tilde{s}')$ gave the readers an impression that we are predicting the actions and states together. We would like to clarify that L287-293 actually states that the semantic transition function predicts $\tilde{s} \rightarrow \tilde{s}'$ (both $\tilde{s}$ and $\tilde{s}'$ are in semantic space) without explicitly predicting the LLM actions. Rather, the "action" for each transition is implicitly recovered as the directional vector $\tilde{a}'=\tilde{s}'-\tilde{s}$ (mentioned in Line 268). In our paper, we don't view or claim this "action"/vector as predicting an LLM response. Rather, we just use these "actions" as part of the MCTS process in the semantic space (e.g., in Algorithm 1 line 7, where we perform action selection and expansion in semantic space). The purpose of the semantic transition model is to predict the approximate transition of conversation semantics (we show that these predictions are reasonable in Sec. A.7) at each step, and does not represent LLM response prediction. Hopefully, this provides some clarification to the author's question.

---

> ### Author Response · Authors · 2024-11-19
>
> ## Part 2/3
>
> > Since SCOPE requires a trained transition and reward function in latent space, it becomes questionable whether SCOPE can generalize when evaluation dialogues become OOD compared to the ones used to train the transition/reward function; or when different LLMs is used to propose candidates at test time. [...] the planning process becomes policy agnostic
>
> To clarify, the transition model we used in our experiments is trained from lmsys data, which comes mostly from conversations between a vicuna-based LLM and a human [3]. On the other hand, during our evaluation, we used Llama-3 as our LLM to generate LLM responses. __Hence, even if a different LLM is used during test time, our method generalizes well and performs better than other baselines__.
>
> In addition, the dialogues we used for evaluation contain a mixture of conversation starters from DailyDialog and lmsys. DailyDialog data is not used to train our transition models. To make our results clearer, we took the evaluation starters which came from the Daily Dialogue dataset and show SCOPE's isolated performance on them. As we see from the table below, _SCOPE outperforms other baselines even though the transition model is trained on a different conversation dataset_ (these results are already part of our paper's empirical results, just that we isolated and show them explicitly here). __Hence, SCOPE still performs well for conversations not explicitly used to train the transition model.__
>
> Length (Higher is better; how much higher than _random_):
>
> | 0-step greedy   | 1-step greedy | SCOPE 2s      | SCOPE 2.5s     | SCOPE 3s      |
> | - | - | - | -| - |
> | \-72 $\pm$ 7.5 |  37 $\pm$ 10 | 122 $\pm$ 12 |  131 $\pm$ 15 | __148 $\pm$ 15__ |
>
> Harmful (Higher is better; how much higher than _random_):
>
> | 0-step greedy   | 1-step greedy | SCOPE 2s      | SCOPE 2.5s     | SCOPE 3s      |
> | - | - | - | -| - |
> | 18 $\pm$ 7.9 |  -11 $\pm$ 14.5 | 29 $\pm$ 7 |  35 $\pm$ 3.9 | __41 $\pm$ 5.1__ |
>
> We hypothesize that SCOPE generalizes well because even though different humans and LLMs converse differently, the stochastic transition of conversation semantics is approximately similar regardless. These are very interesting discussion points, and we would like to incorporate them in the revised paper, thank you.
>
> ---
>
> > Since SCOPE planning is conducted in latent semantic space, there is a lack of transparency/explanability in its decision making process. This is in contrast to approaches that plans in text space (e.g., prompt based MCTS). This could present difficulties to researchers or users to understand how or why certain actions were chosen.
>
> Thank you for the comment. While SCOPE achieves higher conversation rewards by planning in semantic space, we agree planning this way makes it difficult to interpret the decision process. We believe this is not an easy problem and should be left as a future research direction. One practical approach would be to use SCOPE for planning and when the need for interpretation arises, use prompt-based MCTS (with larger time budgets) to verify why a certain action is picked. Alternatively, we could use the encoder from an encoder-decoder model as the semantic embedding model, and using the decoder to interpret predicted states. We will mention this in our revised paper.
>
> ---
>
> > In experiments you used $\lambda=0.1$ for UCT, which forces the tree search to focus on exploitation instead of exploration. This is rather an uncommon value. Is there a reason for this?
>
> We used $\lambda=0.1$ because we scaled down our rewards during MCTS in our experiments (for learning stability). As a result, the predictions for $Q_k(s,a)$ in Equation 3 are relatively small compared to the second term in the equation, and $\lambda=0.1$ was chosen to balance the 2 terms.
> Hence, it was sufficient enough to promote exploration as well. We will improve the writing by mentioning this in the appendix.
>
> ---
>
> > Can you provide more details about the benchmarks you tested? Currently its only mentioned in L363-365 as "dialogue datasets consisting of open conversations and dialogue between LLM and humans". Are these generic dialogues from existing chat datasets or are these curated from certain dialogue planning benchmarks?
>
> Thank you for the question. We used actual conversation starters from the DailyDialog and lmsys dataset (around half of evaluation data  is taken from each dataset, total of 100 starters for each evaluation task). We take the first turn statement for each conversation in those datasets and treat it as the conversation starter. The reason why we used lmsys is because it actually contains conversations between humans and LLMs, and occasionally some harmful topics. DailyDialog conversation is similar to real-world conversations, which we think future LLMs serving as chatbots or companions might encounter. We will improve the clarity of our experimental setup in the revised paper.

---

> ### Author Response · Authors · 2024-11-19
>
> ## Part 3/3
>
> > Other Comments
>
> > Planning in semantic/latent space (L108-111) has been explored in some prior work [1-2]. These should be mentioned in this paper as related work.
>
> Thank you for pointing this out, we will cite [1] and [2] in our revised paper's related work for planning in semantic space. The key difference is that prior works attempt to learn a latent space policy by fine-tuning the language model. On the contrary, our method is more lightweight because it merely approximates the transition of conversation semantics, before using it to infer which LLM response leads to higher reward during inference time, without needing any LLM training.
>
> ---
>
> > Currently Introduction and Background/Related work takes up more than 4 pages. This is too long, as it leaves little room for methods and experiments. I would suggest the authors to trim Section 1-3 as much as possible (e.g., details about MCTS can be moved to appendix).
>
> Thank you for the suggestion, we will try our best to trim the section. We felt that the details of MCTS is important in this paper (as opposed of just moving them entirely to appendix) because 1) it serves as a backbone to our algorithm and 2) after we project the conversations into semantic space, we want to explain precisely which part of MCTS stays the same and which part is now different.
>
> ---
>
> > If I understood correctly, "0-step greedy" directly chooses the best response according to the reward model? If so, this should be named "rejection sampling" instead, which is a common approach used in many RL related work.
>
> We chose the method name as 0-step greedy to mark out the clear distinction between greedy and non-myopic approaches. However, we are aware that some RL related work might have used rejection sampling to denote such methods. We will make a note in the revised paper that some literature might use a different name for such methods, to make it clearer for readers from different backgrounds.
>
> ---
> > In L259 and L346, it should be "conversation states s" instead of "conversation starter s"
>
> > "Section 6.5 Conclusion" should be "Section 7 Conclusion".
>
> Thank you for the comments on our paper's writing. We will incorporate the reviewer's suggestion on related works and improve the writing in our revised paper.
>
> ---
>
> Once again, we sincerely hope that our additional experimental results and clarifications have addressed your questions satisfactorily and improved your opinion on our work. If you are satisfied with the discussion, we will incorporate our responses and clarifications into the revised paper.
>
> [1] Lubis, Nurul et al. “LAVA: Latent Action Spaces via Variational Auto-encoding for Dialogue Policy Optimization.” ArXiv abs/2011.09378 (2020): n. pag.
>
> [2] Vlastelica, Marin et al. “Taming Continuous Posteriors for Latent Variational Dialogue Policies.” AAAI Conference on Artificial Intelligence (2022).
>
> [3] Zheng et al. (2023) LMSYS-Chat-1M: A Large-Scale Real-World LLM Conversation Dataset

---

> ### Comment · Reviewer_Cm3A · 2024-11-19
>
> I would like to follow up on the following question, which is my main technical concern about this work.
>
> > This work trains a transition function to predict $T(s) \to (a',s')$ instead of $T(s,a') \to s'$, based on description in L287-293. This means that this transition function needs to predict both the response that will be generated by the LLM and next the corresponding user response. This seems unrealistic because 1) if it can accurately model then it essentially becomes an LLM, and 2) the planning process becomes policy agnostic (also see Algorithm 1 line 7) - a sign indicating that SCOPE may not be robust against using different LLMs as policy models (unlike prompt based MCTS).
>
> Thank you for your response. My point here is that 1) a dialogue state $s$ essentially means a sequence of (user text, LLM text, user text, LLM text, ...), and that 2) going from a a dialogue state $s$ to the next one $s'$ only make sense if you have access to the next LLM text and a (simulated) user text.
>
> You mentioned the semantic transition function directly predicts $s \to s'$. To my point above, I believe this already means that it has to model the next LLM text and user text (semantically). Even though "In our paper, we don't view or claim this "action"/vector as predicting an LLM response", I believe modeling $s \to s'$ is implicitly doing this. Otherwise how can you model transitioning from a dialogue state to the next one?
>
> Please let me know if I misunderstood anything.

---

> > ### Author Response · Authors · 2024-11-20
> >
> > Thank you for engaging in further discussions with us. Just to clarify, in our work, the semantic transition function models the conversation semantic transitions $\tilde{s}$ to $\tilde{s}'$ ($\tilde{s}$ and $\tilde{s}'$ are points in the semantic space and we perform MCTS entirely in this semantic space during planning) rather than simulating textual conversations $s$ to $s'$ (which is what prompt-based MCTS does). By doing so, the semantics of the LLM action is captured in the semantic transition model's predictions.
> >
> > We agree that modeling the transition of conversation semantics appears, in certain aspects, similar to implicitly modeling the LLM response (e.g., the reviewer remarked that "if it (transition model) can accurately model $a'$, then it essentially becomes an LLM"). However, modeling the conversation semantic transitions $\tilde{s} → \tilde{s}'$ (in our paper) is a much simpler task than predicting LLM responses directly. There are two key difference in the approaches. First, predicting the LLM and user response directly (e.g., token by token using another LLM, like what prompt-based MCTS does) is much more compute-intensive than predicting the transition in conversation semantics. Second, we do not need to model the semantic transitions of LLMs and humans responses precisely, but rather just well enough for us to estimate the rewards associated to each starting action (LLM response) to preserve the ranking of actions after performing MCTS solely in semantic space. For example, even though there are some prediction errors associated with the semantic transition model (as seen in Sec. A.7), SCOPE still attains higher rewards after planning.
> >
> > As a result, our paper shows that the resulting semantic transition model is lightweight and efficient to use, incurring much less MCTS search time in semantic space than conventional prompt-based MCTS (lines 61-75) and achieving better performance.
> >
> > >  the planning process becomes policy agnostic [...] may not be robust against using different LLMs as policy models.
> >
> > Our paper's empirical results show that SCOPE does work well even when used with a different LLM (policy model). The semantic transition model we used in our experiments is trained from lmsys data, which comes mostly from conversations between vicuna-based LLM and human users [1]. On the other hand, in our paper's evaluation, we used Llama-3 as the LLM to generate LLM candidate responses. Hence, even if a different LLM is used during test time, our method generalizes well and performs better than other baselines.
> >
> > We hypothesize this occurs because even though different LLMs speak differently, the semantic content of their responses are approximately similar. Therefore, our semantic transition model can generalize well to different LLMs. This enables SCOPE to perform well even though a different LLM is used during test time.
> >
> > Additionally, for actual deployment of SCOPE in the real world, the LLM provider could opt to train the transition models with actual LLM user conversations collected from deployment. This would align the transition model better with the specific LLM model, thereby achieving better performance when performing conversation planning with SCOPE.
> >
> > We thank the reviewer for these fruitful discussion and hope our responses have clarified your questions and improved the opinion of our work.
> >
> > [1] Zheng et al. (2023) LMSYS-Chat-1M: A Large-Scale Real-World LLM Conversation Dataset

---

> > > ### Comment · Reviewer_Cm3A · 2024-11-20
> > >
> > > Thank you for your replies. I agree that the overall positive empirical results shows that SCOPE can improve an LLM's response quality. However, it does not contradict my concern about the underlying methods, and its potential limitations. I believe this is an interesting work, and would like to keep my score of 6 based on the responses.
> > >
> > > ---
> > >
> > > I summarize some of the main concerns I have below
> > >
> > > - The trained transition function to model dialogue state transitions practically has to model the next LLM text and user text (semantically). The authors argue that "modeling the conversation semantic transitions (in our paper) is a much simpler task than predicting LLM responses directly". Is there any empirical evidence/prior work backing this? Please let me know if I missed it.
> > >
> > > - Since the planning process of SCOPE is *policy agnostic*, this is essentially like "off-policy" planning. While I agree that SCOPE does show improved results in the experiments in paper, I believe this direction has strong limitations. For example, since the transition model only takes in a prior dialogue state $s$, SCOPE should return **identical simulation results** when a) a weak GPT-2 model will be used to generate next response; and b) models like GPT-4o will be used.

---

> > > > ### Author Response · Authors · 2024-11-25
> > > >
> > > > Thank you for your prompt replies! At this point, it is unclear how an "on-policy" approach would differ from our approach in terms of results and compute/resource requirements (see below for some minor clarifications). In general, extending SCOPE to "on-policy" planning would be a promising future direction and our approach can serve as a competitive baseline and foundation for future works on on-policy approaches in terms of performance-efficiency trade off.
> > > >
> > > > Finally, we would like to make some minor clarification of the reviewer's closing statements:
> > > >
> > > > > The authors argue that "modeling the conversation semantic transitions (in our paper) is a much simpler task than predicting LLM responses directly"
> > > >
> > > > By "simpler task", we meant that, in our context, we were able to model conversation semantic transitions reasonably well with a lightweight model, which allows us to perform MCTS much faster as compared to using an LLM (e.g., prompt-based MCTS) under practical & tight planning budgets.
> > > >
> > > > > since the transition model only takes in a prior dialogue state $s$, SCOPE should return identical simulation results when a) a weak GPT-2 model will be used to generate next response; and b) models like GPT-4o will be used.
> > > >
> > > > We want to make a minor clarification: SCOPE uses the LLM to propose a candidate set of responses and uses them for the first level of action expansion from starting dialogue state during MCTS (Line 2 & 3 of Algo. 1). As different LLMs would typically propose different starting candidate responses even with the same prior dialogue states, we would have started from different points in semantic space early on and hence the simulation results will be different.
> > > >
> > > > We will incorporate the valuable feedback into our revised paper. Thanks again for your time and reviews.

---

> > > > > ### Comment · Reviewer_Cm3A · 2024-11-25
> > > > >
> > > > > > since the transition model only takes in a prior dialogue state, SCOPE should return identical simulation results when a) a weak GPT-2 model will be used to generate next response; and b) models like GPT-4o will be used.
> > > > >
> > > > > I am quite familiar with MCTS, but maybe I misunderstood your Algorithm 1. Given an initial dialogue state (e.g., *a user query only*), you have state $s_{init}$ as mentioned in Algo 1 Line 1-3. Then, in your first iteration of node expansion (Line 7), you need to call your transition model $\tilde{T}$ to obtain a new state. Since in this work you consider $\tilde{T}(s) \to s'$, Line 7 would result in new nodes/next state being *identical regardless of what your policy model is*?
> > > > >
> > > > > I agree that as more and more context are added in the state (i.e., when there are multiple LM responses in your states already), the chance that states using different policy model reaches (semantically) similar states is low (*albeit not impossible*). However, I believe there are fundamental issues when transition is modeling as $\tilde{T}(s) \to s'$, i.e., it becomes off-policy as mentioned in the previous response.

---

> > > > > > ### Author Response · Authors · 2024-11-27
> > > > > >
> > > > > > _(We have updated a revised version of our paper (changes in blue) which has incorporated the reviewer's prior concerns and some clarifications.)_
> > > > > >
> > > > > > We'd like to first clarify our algorithm: in SCOPE, our transition model does not predict $\tilde{s}$ to $\tilde{s}'$ directly in one single step. Instead, it first predicts $\tilde{a}$ from $\tilde{s}$, the semantic representation of the LLM response $a$ to the conversation context $s$ (Figure 9 (b)). Secondly, it predicts $\tilde{s}'$ given $(\tilde{s},\tilde{a})$, the transition of conversation semantics to the next state $s'$ after the human user responds (Figure 9 (c)).
> > > > > > So, each transition step comprises of two intermediate movements in semantic space.
> > > > > >
> > > > > > To give a concrete example: if a conversation contains the following content (A,B,C represents texts):
> > > > > > - Human: A (starting state $s$)
> > > > > > - LLM: B (action $a$)
> > > > > > - Human: C (A,B,C together represents $s'$)
> > > > > >
> > > > > > Our transition model predicts (__semantically__) how the conversation would first change from (A) to (A,B), and then change from (A,B) to (A,B,C). By doing these two steps, we have predicted $\tilde{s} \rightarrow \tilde{s}'$  (A to A,B,C in semantic space). In our paper, the transition model comprises of two sub-models that performs each of the two steps. These models are trained in a similar fashion, as illustrated in Appendix A.7. We have added more details in Appendix A.5 to elaborate on the training procedure. Training two sub-models to predict $\tilde{a}$ from $\tilde{s}$, and $\tilde{s}'$ from $(\tilde{s},\tilde{a})$ respectively has the following advantages:
> > > > > >
> > > > > > 1. Doing so aligns SCOPE with the steps in the MCTS framework, which requires us to select actions at each node and generate new states from node expansion (this is exactly equals to the two steps mentioned above) in line 7 of Algo 1, where we "sample actions" and "sample new states from the selected actions" (in semantic space).
> > > > > > 2. We can directly use the initial LLM candidate responses as actions at the root node of MCTS instead of predicting the LLM responses semantically using our transition model (this should address your question in the next part), so the node expansion of $\tilde{T}(\tilde{s},\tilde{a}) → \tilde{s}'$ from the root node will be not be the same if different LLMs and responses (different $\tilde{a}$) are used.
> > > > > >
> > > > > > We apologize if our previous comments gave the impression that we are doing the prediction directly _in one step_. We have added additional explanation in Section 5 and Appendix A.5 in the revised paper to make this point clearer.
> > > > > >
> > > > > > There might be some concern whether our transition model is essentially training another LLM (you mentioned this in the first review), but as we have mentioned in our previous response, our goal is to predict the change in conversation semantics and we did so quite successfully using a lightweight and compute efficient model in our paper.
> > > > > >
> > > > > > > I am quite familiar with MCTS, but maybe I misunderstood your Algorithm 1. Given an initial dialogue state (e.g., a user query only), you have state $s_{init}$
> > > > > >  as mentioned in Algo 1 Line 1-3. Then, in your first iteration of node expansion (Line 7), you need to call your transition model $\tilde{T}$
> > > > > >  to obtain a new state. Since in this work you consider $\tilde{T}(s) \rightarrow s'$
> > > > > > , Line 7 would result in new nodes/next state being identical regardless of what your policy model is?
> > > > > >
> > > > > > To address your last question, in line 7 of Algo 1 the transition to $\tilde{s}'$ in semantic space is dependent on the semantic action $\tilde{a}$ that we select in line 6 under the current node. For the root node, this semantic action comes from the initial set of LLM candidate responses in which we perform SCOPE on (to select the best one). In this case the new nodes/next state $\tilde{s}'$ from doing node expansion $\tilde{T}(\tilde{s},\tilde{a}) → \tilde{s}'$ will not be the same since the initial set of candidate actions in semantic space, $\tilde{a}$, is different. Hope this answers your question on why the simulation result will not be identical.
> > > > > >
> > > > > > Lastly, our response here does _not_ invalidate your points regarding on-policy vs off-policy (__we are merely trying to make a clarification regarding the earlier question on whether simulation results are identical with different LLMs__). In fact, we agree with you that adopting SCOPE to an on-policy variant is a challenging and important future work (possibly with room for improvement), and we have acknowledged this limitation in our paper and responses (Conclusion and Appendix A.5 in the revised paper). Thanks again for helping us to improve our paper's presentation and content!

---

> ### Comment · Reviewer_Cm3A · 2024-11-27
>
> Thank you for the clarification. Before proceeding with my concern, I want to clarify my stance. I provided an overall positive score for this work, since I believe performing MCTS in a semantic/hidden space can speed up its inference time and is of practical value. However, as there is often no free lunch, the proposed method has limitations, i.e., requires transition model to be also in the semantic space without additional query to the language model for next actions (hence my comment on off-policy and model agnostic).
>
> I feel like the authors are trying to address the "model agnostic" example I provided, but I note that I only mentioned it as an example to clarify my concern. I agree with the "fix" the author mentioned can be done to avoid that particular case, but I believe this misses my main point. I re-emphasize them below.
>
> ---
>
> > I am quite familiar with MCTS, but maybe I misunderstood your Algorithm 1. Given an initial dialogue state (e.g., *a user query only*), you have state $s_{init}$ as mentioned in Algo 1 Line 1-3. Then, in your first iteration of node expansion (Line 7), you need to call your transition model $\tilde{T}$ to obtain a new state. Since in this work you consider $\tilde{T}(s) \to s'$, Line 7 would result in new nodes/next state being *identical regardless of what your policy model is*?
>
> - "We'd like to first clarify our algorithm: in SCOPE, our transition model does not predict $\tilde{s}$ to $\tilde{s}'$ directly in one single step."
>
>   \
>   I note that the essence of my concern is that the *only physical input your $\tilde{T}$ relies on is only a state*, and its final output is the next state. This means future simulation is off-policy and model agnostic. It does not matter *how you predict the next state*, but rather there is an **information bottleneck** where your transition model does not query the language model policy for future actions and only relies on current states (i.e., dialogue history). This is in contrast to the other MCTS methods I mentioned that plans in text space, where to model future states they also query the language policy for its actual future actions to model transitions.
>
> - "we are merely trying to make a clarification regarding the earlier question on whether simulation results are identical with different LLMs"
>
>    \
>    The proposed algorithm proposed conducts simulation without relying on querying LLM for future responses, and practically the only place it queries the LLM is in the beginning (L2 in Algo 1). *In general*, I believe for your statement to strictly hold true there are two assumptions. 1) Different LLMs is *guaranteed* to provide distinct responses in L2 in Algo 1, and that 2) even when you received slightly different responses, they have to be different enough in semantic space such that *all* of your subsequent simulation reaches different outcomes (that are hopefully faithful to the actual LLM's behavior). There is also a concern whether or not your trained transition model can really differentiate semantic transitions for different policies when the only place the LLM is actually used is L2, but I believe this is an empirical question and is less of a concern given the generally positive results for the proposed method.
>
>    \
>    My emphasis here is that these are not concerns for on-policy MCTS that queries the LLM during simulation, and are additional requirements for the proposed method to work well *fundamentally due to the information bottleneck I mentioned in my previous point*.

---

> > ### Author Response · Authors · 2024-11-29
> >
> > Thank you for the positive score and noting the practical value of our work. The discussion has been very fruitful and insightful, and we understand your point. Thanks again for reviewing our paper!

---

### Official Review · Reviewer_5qmX · 2024-10-30

**Soundness:** 2
**Presentation:** 3
**Contribution:** 3
**Rating:** 8
**Confidence:** 4

**Summary:**

The authors propose a method called SCOPE (Semantic space COnversation Planning with improved Efficiency) which focuses on making conversation planning in LLMs more efficient. There is a need to look ahead at every point in the conversation to see if choosing a particular response will lead to a better conversation in the long run; however, the authors mention that current methods that use vanilla MCTS are time-consuming because an LLM will need to be queried multiple times to get all possible scenarios.

Therefore the authors propose a method that doesn't involve querying an LLM when determining future states but rather leverage the semantic space for more efficient searching. More specifically SCOPE involves 1) training a Transition model that samples a state where a state is a conversation context ending in a Human Turn and 2) training a reward model that predicts the reward at a certain state. The reward is the number of tokens in the user output and harmlessness which is predicted from Llama-Guard 2. To project the conversation and response into a semantic space the authors use the feature layer of Llama Guard 2 as the semantic embedding.

The authors then compare their SCOPE method against a variety of baselines which include: not doing any conversation planning, doing conversation planning for only one step, vanilla MCTS (which is time-consuming) and selecting a random response. They evaluate by measuring the cumulative reward and find that SCOPE outperforms all these methods and is much more efficient than vanilla MCTS. Both the training and testing were done on the Lmsys-chat-1m and Daily Dialog datasets.

They ran ablation studies to find what is the best type of model architecture to use for their Transition model and how many turns is good enough to plan ahead.

**Strengths:**

1) The goal of this paper is well motivated. Working towards a more efficient conversation planning method can help with customer experience since latency will decrease and it seems the proposed method is novel. I think this will further encourage future work in this area.

2) The paper is well-written and easy to follow. I appreciate diagrams such as Figure 8 which helped visualize their overall Algorithm. Additionally the explanation of their method is also clear and easy to follow. In addition to giving good details on their experimentation the authors also released their code which will make it useful for the community to reproduce and build off of.

**Weaknesses:**

EDIT AFTER AUTHOR RESPONSE: I am satisfied with both answers from the authors regarding the reward model and evaluation. I believe my score is already high so will be keeping it as 8.

My biggest concern is around the evaluation of this method along with the reward model.

Regarding the reward model: I think that the harmlessness metric makes sense and the use of Llama-Guard2 is a good decision. However for engagement I don't think just measuring the token length of the user response is enough. Yes that is definitely a fine proxy to have but I don't think it is enough and I don't think "greater commercial benefits" is a good enough motivation. For one thing if this method was say used in spoken conversations then token length wouldn't be a good enough metric. One idea is to perhaps measure how often is the user asking questions to show that they are engaged in the conversation.

Regarding evaluation: Overall the authors look at maximizing the cumulative reward to determine what is the best method in this case which is a good setup but I would think having some human evaluation could help solidify their arguments unless they disagree in which case I'm happy to hear why.

**Questions:**

As mentioned above if the authors can address my concern regarding evaluation and the choice of reward functions.

Another question I have is do you think the transition model you trained on those datasets will generalize to other unseen domains? Has this been looked at?

---

> ### Author Response · Authors · 2024-11-19
>
> ## Part 1/2
>
> We would like to thank the reviewer for the comprehensive review and compliments for our method's motivation, presentation and reproducibility. We would like to present our response to the reviewer's questions below.
>
> ---
>
> > Regarding the reward model: I think that the harmlessness metric makes sense and the use of Llama-Guard2 is a good decision. However for engagement I don't think just measuring the token length of the user response is enough. Yes that is definitely a fine proxy to have but I don't think it is enough [...] One idea is to perhaps measure how often is the user asking questions to show that they are engaged in the conversation.
>
> Thank you for the question. We would like to emphasize that our method, SCOPE, is reward agnostic and one can use any reward function in SCOPE to plan. We agree with the reviewer that there might be other conversation engagement metrics that works better as the reward function. However, the actual choice of reward function is not the focus of our work - our main contribution (like what the reviewer has pointed out) is an efficient way to do conversation planning to maximize a reward function in the conversation MDP setting and our evaluation is done on two reward functions that, as the reviewer pointed out, serves as a good proxy for certain real-world applications (below, we provided more experimental results on another engagement evaluation metric). Hence, the best reward function depends heavily on the problem setting and the LLM owner's preference. We hope our paper can inspire people to try using SCOPE with other reward functions. We will re-emphasize this point in our revised paper.
>
> ---
>
> > Overall the authors look at maximizing the cumulative reward to determine what is the best method in this case which is a good setup but I would think having some human evaluation could help solidify their arguments unless they disagree in which case I'm happy to hear why.
>
> Thank you for the suggestion. We agree that human evaluation is the gold standard to evaluate conversations. For example, LLM owners can deploy SCOPE and use human annotators to rate whether more engaging conversations are produced. Unfortunately at this point of time, we do not have the time budget to conduct full-fledged human evaluation. Despite so, we believe cumulative rewards, as the reviewer put it, serve as a good proxy for an evaluation metric. To make our evaluation stronger, we adopted the reviewer's advice of measuring "how often the user is asking questions to show that they are engaged in the conversation" to check if SCOPE indeed produces more engaging conversations (we still use cumulative rewards for planning, but during evaluation, we check if the user is asking questions in the resulting conversation). The results show that in conversations produced by SCOPE, the user on average asks more questions, signaling that they are more engaged with the conversation.
>
> | random   | 1-step greedy | 0-step greedy   | SCOPE 3s     |
> | --------------- | ------------- | ------------- | -------------- |
> | 2.99 |  2.95 | 2.97 |  __3.31__ |
>
> We will include the additional experiments and add some discussion regarding human evaluation in our revised paper.

---

> > ### Comment · Reviewer_5qmX · 2024-11-20
> > **Addressing Evaluation**
> >
> > Thank you for addressing my questions I'm satisfied with the answers. It's clear that the focus of the work isn't specifically on the reward model and the current scope (pun not intended) is on the planning. Future work could involve adding other metrics for the reward model which could be interesting.

---

> > > ### Author Response · Authors · 2024-11-25
> > >
> > > Thanks for the valuable feedback, we will incorporate the helpful suggestions into our updated paper.

---

> ### Author Response · Authors · 2024-11-19
>
> ## Part 2/2
>
> > do you think the transition model you trained on those datasets will generalize to other unseen domains? Has this been looked at?
>
> We trained our transition model with the lmsys dataset but in our experiments, we also evaluated SCOPE on conversation starters from the Daily Dialog dataset [1] (see line 1057 for detailed experimental setup) which contains different dialogues as compared to lmsys dataset. Furthermore, the lmsys dataset contains mostly dialogues from the vicuna model, while our experiments were evaluated on the Llama-3 model. Therefore, it can be seen that SCOPE indeed generalizes to conversation contexts outside of the data used to train our transition model, achieving higher cumulative rewards.
>
> To make our results clearer, we took the evaluation starters which came from the Daily Dialogue dataset and show SCOPE's isolated performance on them. As we see from the table below, SCOPE outperforms other baselines even though the transition model is trained on a different conversation dataset (this result is already part of the empirical results in Fig. 3 of our paper, just that we show it in isolation here).
>
> Length (how much longer was human responses in the conversation as compared to _random_)
>
> | 0-step greedy   | 1-step greedy | SCOPE 2s      | SCOPE 2.5s     | SCOPE 3s      |
> | --------------- | ------------- | ------------- | -------------- | ------------- |
> | \-72 $\pm$ 7.5 |  37 $\pm$ 10 | 122 $\pm$ 12 |  131 $\pm$ 15 | __148 $\pm$ 15__ |
>
> Harmful (how much less harmful (Llama-guard) was the conversation as compared to _random_)
>
> | 0-step greedy   | 1-step greedy | SCOPE 2s      | SCOPE 2.5s     | SCOPE 3s      |
> | --------------- | ------------- | ------------- | -------------- | ------------- |
> | 18 $\pm$ 7.9 |  -11 $\pm$ 14.5 | 29 $\pm$ 7 |  35 $\pm$ 3.9 | __41 $\pm$ 5.1__ |
>
> In real-world settings, an LLM owner can also use a user's data (if they permit) to fine-tune the transition models to match the user's demographic and speaking pattern, possibly improving SCOPE's effectiveness even further. This will be an interesting future research direction, which we will mention in our revised paper.
>
> ---
>
> We sincerely hope that our additional experimental results and clarifications have addressed your questions satisfactorily and can improve your opinion of our work. If you are satisfied with the discussion, we would incorporate our responses into the revised paper.
>
> [1] Li, et al. (2017). DailyDialog: A Manually Labelled Multi-turn Dialogue Dataset.

---

### Official Review · Reviewer_o97T · 2024-11-03

**Soundness:** 3
**Presentation:** 2
**Contribution:** 4
**Rating:** 8
**Confidence:** 4

**Summary:**

This paper introduces SCOPE, a novel approach for efficient conversation planning with LLMs. It leverages dense semantic representations to model stochastic transitions and rewards within conversations, enabling faster planning without additional LLM queries. SCOPE achieves higher cumulative rewards compared to conventional simulation-based planning algorithms, demonstrating its effectiveness in real-time conversations.

**Strengths:**

The main innovations of this paper:
Introduces the concept of representing conversations in a dense semantic space, which captures the semantics of natural language conversations effectively. This representation allows for the modeling of stochastic transitions within a conversation and their associated rewards. Compare with the language or token space, this method helps to achieve a significant improvement in planning speed and improves the diversity of LLM samples.

**Weaknesses:**

The paper relies on a specific dataset (1msys/1msys-chat-1m) for training the transition models. It would be beneficial to demonstrate the generalizability of SCOPE by testing it on additional datasets or in different conversational contexts.

**Questions:**

How does SCOPE handle the potential bias introduced by the semantic embedding model？

---

> ### Author Response · Authors · 2024-11-19
>
> We would like to thank the reviewer for the comprehensive review and compliments for our method's innovation and contribution in making conversation planning faster.
>
> ---
>
> To address the reviewer's comments, we have provided additional clarifications. We hope you find them enlightening and useful.
>
> > The paper relies on a specific dataset (1msys/1msys-chat-1m) for training the transition models. It would be beneficial to demonstrate the generalizability of SCOPE by testing it on additional datasets or in different conversational contexts.
>
> Even though we trained our transition model with the lmsys dataset, in our experiments, the set of conversation starters that SCOPE is evaluated on contains starters from the Daily Dialogue dataset [1] (line 1057 for detailed experimental setup), which has different conversation starters from the lmsys dataset. Therefore, it can be seen that SCOPE indeed generalizes to conversations outside of the data used to train our transition model, achieving higher cumulative rewards.
>
> To make our results clearer, we took the evaluation starters which came from the Daily Dialogue dataset and show SCOPE's isolated performance on them. As we see from the table below, _SCOPE outperforms other baselines even though the transition model is trained on a different conversation dataset_ (this result is already part of our paper's empirical results, just that we show it in isolation here).
>
> Length (Higher is better; how much higher than random):
> | 0-step greedy   | 1-step greedy | SCOPE 2s      | SCOPE 2.5s     | SCOPE 3s      |
> | --------------- | ------------- | ------------- | -------------- | ------------- |
> | \-72 $\pm$ 7.5 |  37 $\pm$ 10 | 122 $\pm$ 12 |  131 $\pm$ 15 | __148 $\pm$ 15__ |
>
> Harmless score (higher is better; how much higher than random):
> | 0-step greedy   | 1-step greedy | SCOPE 2s      | SCOPE 2.5s     | SCOPE 3s      |
> | --------------- | ------------- | ------------- | -------------- | ------------- |
> | 18 $\pm$ 7.9 |  -11 $\pm$ 14.5 | 29 $\pm$ 7 |  35 $\pm$ 3.9 | __41 $\pm$ 5.1__ |
>
> In real-world settings, an LLM owner can also use a user's data (if they permit) to fine-tune the transition models to match the user's demographic and speaking pattern, possibly improving SCOPE's effectiveness even further. This will be an interesting future research direction, which we will mention in our revised paper.
>
> ---
>
> > How does SCOPE handle the potential bias introduced by the semantic embedding model？
>
> Thanks for the interesting question. As we pointed out in Section A.8 (How does transition and reward model performance affect SCOPE?) and our experiments, even when our semantic embedding or transition model has some inaccuracies, our empirical results have shown that SCOPE still achieves higher cumulative rewards than other methods. There could be a few explanation for this. __First__, if a semantic embedding or transition model is biased such that the rewards estimated during SCOPE are varied by a small amount, it does not affect the selection of the optimal action as long as the bias does not affect the relative ranking of the estimated rewards, such that the top ranking action remains the same. __Second__, even if there are errors in the models, because SCOPE is able to perform so many more rounds of MCTS rollouts (92 times more than vanilla MCTS, according to section A.9) within a short amount of time, it can still estimate the rewards associated with each possible LLM response more accurately than conventional MCTS (which uses LLM simulation) that has large sampling error due to insufficient number of rollouts within a tight planning budget.
>
> Once again, thank you for the positive feedback and comments. We sincerely hope that our clarifications have addressed your questions satisfactorily and can improve your opinion of our work. If you are satisfied with the discussion, we would incorporate our responses above to improve our paper's writing and clarity.
>
> [1] Li, et al. (2017). DailyDialog: A Manually Labelled Multi-turn Dialogue Dataset.

---

> > ### Author Response · Authors · 2024-12-02
> >
> > Once again, thanks for reviewing our paper! Do let us know if you have any other questions and we will be happy to address them.

---

### Author Response · Authors · 2024-12-02

We would like to thank the reviewers for the reviews, positive comments and scores, specifically:
- SCOPE's novel use of dense semantic space to model transition and rewards for conversation planning in a light-weight fashion (all reviewers).
- SCOPE's ability to improve planning speed (more than 70x in our paper) in conversations without relying on expensive LLM queries (all reviewers).
- Our practical problem setting of conversation planning in areas such as customer experience (reviewer `5qmX`).
- Our paper's well-written presentation and figures (reviewer `5qmX`).
- Our paper's code reproducibility and well thought out empirical evidence/theoretical analysis (reviewers `5qmX`, `Cm3A`).

We would like to summarize our main responses and discussions during the rebuttal period.

- We clarified with specific empirical results that our approach is able to generalize to conversation starters that are not present in the transition model training data (all reviewers)
- We highlighted the validity of our reward function choices used in the paper and discussed evaluation metric alternatives that could be used in real-world settings (reviewer `5qmX`).
- We had a fruitful discussion about the merits of SCOPE as compared to other approaches such as prompt-based MCTS ("on-policy"). We noted that without real-time requirements or with significant runtime compute improvements, it's possible that other on-policy approaches in the future can potentially trade off time efficiency for better performance. Currently, our paper showed that SCOPE empirically achieves better performance with smaller runtime than other existing approaches, including on-policy ones like prompt-based MCTS. We also noted possible extension of SCOPE (e.g., fitting the transition model with conversation data produced by the specific LLM used during deployment) to address the "model-agnostic" point that reviewer `Cm3A` raised.
- We have incorporated writing suggestions and feedback from the reviewers in our revised paper to improve its presentation (changes in _blue_).

We appreciate the reviewers' time with us and sincerely hope SCOPE serves as a competitive baseline and foundation for future works on other possible conversation planning approaches in terms of performance-efficiency trade-off.

Best regards,

Authors

---

### Meta-Review · Area_Chair_A4Lh · 2024-12-19

**Metareview:**

The authors propose a novel approach to conversation planning, called SCOPE, that addresses the time-consuming nature of existing simulation-based methods. SCOPE models stochastic transitions and rewards within the semantic space of conversations, enabling efficient selection of optimal LLM responses without requiring additional LLM queries for simulation. The reviewers appreciate the novelty of the approach, the evaluation and empirical results, and the overall clarity of presentation. However, they also raise some concerns, such as reliance on specific datasets (and how to generalize), limitations of the reward model, the "training-free" claim might not hold, and lack of explainability in SCOPE's decision making process. The authors provide detailed responses to all concerns and questions.

**Additional Comments On Reviewer Discussion:**

The authors provide detailed responses to all concerns and questions that the reviewers raise, conducting additional experiments where necessary. Regarding the discussions, one reviewer did not engage in discussions, one was satisfied with the response, and one engaged in a lengthier discussion on SCOPE's transition model and on- / off-policy training. The authors also provide a summary of their discussions in a separate post. Overall I find their responses convincing and agree with the reviewers.

---

### Decision · Program_Chairs · 2025-01-22

Accept (Spotlight)